# CRISPR-Cas12a exploits R-loop asymmetry to form double-strand breaks

**Joshua C Cofsky[1], Deepti Karandur[1,2,3], Carolyn J Huang[1], Isaac P Witte[1], John Kuriyan[1,2,3,4,5], Jennifer A Doudna[1,2,3,4,5,6,7]\***

[1]Department of Molecular and Cell Biology, University of California, Berkeley, Berkeley, United States; [2]California Institute for Quantitative Biosciences (QB3), University of California, Berkeley, Berkeley, United States; [3]Howard Hughes Medical Institute, University of California, Berkeley, Berkeley, United States; [4]Department of Chemistry, University of California, Berkeley, Berkeley, United States; [5]MBIB Division, Lawrence Berkeley National Laboratory, Berkeley, United States; [6]Innovative Genomics Institute, University of California, Berkeley, Berkeley, United States; [7]Gladstone Institutes, University of California, San Francisco, San Francisco, United States

**Abstract** Type V CRISPR-Cas interference proteins use a single RuvC active site to make RNA-guided breaks in double-stranded DNA substrates, an activity essential for both bacterial immunity and genome editing. The best-studied of these enzymes, Cas12a, initiates DNA cutting by forming a 20-nucleotide R-loop in which the guide RNA displaces one strand of a double-helical DNA substrate, positioning the DNase active site for first-strand cleavage. However, crystal structures and biochemical data have not explained how the second strand is cut to complete the double-strand break. Here, we detect intrinsic instability in DNA flanking the RNA-3′ side of R-loops, which Cas12a can exploit to expose second-strand DNA for cutting. Interestingly, DNA flanking the RNA-5′ side of R-loops is not intrinsically unstable. This asymmetry in R-loop structure may explain the uniformity of guide RNA architecture and the single-active-site cleavage mechanism that are fundamental features of all type V CRISPR-Cas systems.

**\*For correspondence:** doudna@berkeley.edu

## Introduction

CRISPR-Cas systems (clustered regularly interspaced short palindromic repeats, CRISPR-associated proteins) provide antiviral immunity to prokaryotes through the RNA-guided nuclease activity of enzymes including Cas9 and Cas12a (*Barrangou et al., 2007*; *Jinek et al., 2012*; *Zetsche et al., 2015*; *Makarova et al., 2015*), which are widely used for programmable genome editing (*Pickar-Oliver and Gersbach, 2019*). Both Cas9 and Cas12a use CRISPR RNA (crRNA) to recognize matching double-stranded DNA (dsDNA) sequences by forming an R-loop structure in which 20 nucleotides (nts) of the crRNA (the crRNA 'spacer') base pair with one strand of the target DNA (*Figure 1A, B*; *Jinek et al., 2012*; *Jiang et al., 2016*; *Zetsche et al., 2015*; *Swarts et al., 2017*). In addition, both protein families must bind to a protospacer-adjacent motif (PAM), a short DNA sequence next to the crRNA-complementary sequence, to initiate R-loop formation (*Bolotin et al., 2005*; *Mojica et al., 2009*; *Sternberg et al., 2014*; *Singh et al., 2018*).

Despite their functional similarities, Cas9 and Cas12a evolved independently (*Koonin et al., 2017*) and use distinct mechanisms of DNA cleavage. Cas9 employs two active sites to generate a blunt DNA double-strand break near the PAM (*Jinek et al., 2012*; *Gasiunas et al., 2012*). In contrast, Cas12a uses a single active site to make staggered cuts distant from the PAM, and the same active site can cleave free single-stranded DNA (ssDNA) non-specifically once the enzyme has been

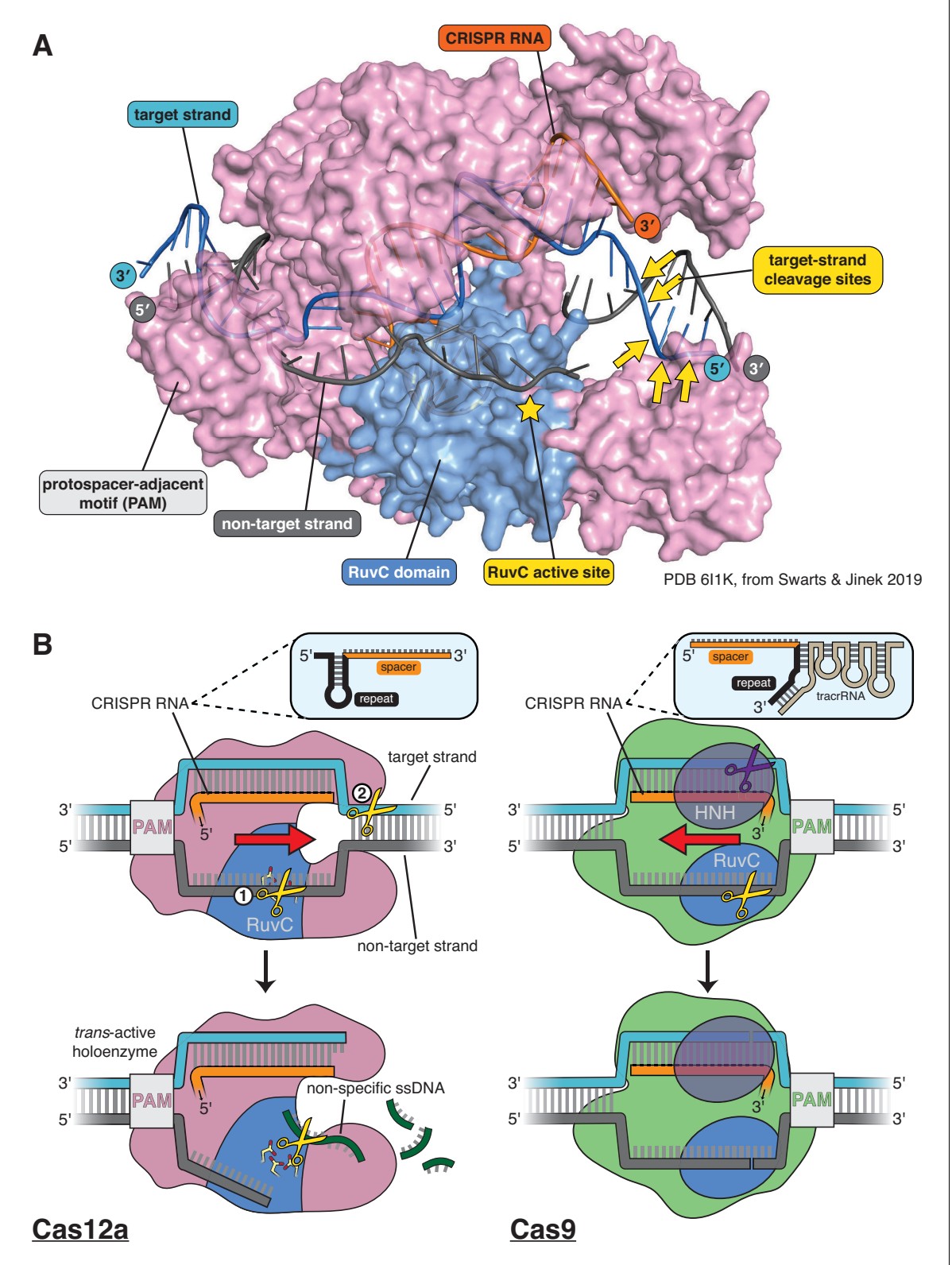

**Figure 1.** Structure of Cas12a and comparison of its DNA cleavage pathway to that of Cas9. (**A**) Crystal structure of the DNA-bound Cas12a interference complex from *Francisella novicida* (FnCas12a, PDB 6I1K) (*Swarts and Jinek, 2019*). While the protein ortholog used for most experiments in this manuscript is from *Acidaminoccus species* (AsCas12a,~40% identity to FnCas12a), the FnCas12a crystal structure shown here represents the most complete structure of such a complex to date, most notably with respect to the DNA at the target-strand cleavage sites. We did not perform any

*Figure 1 continued on next page*

*Figure 1 continued*

experiments with the particular DNA sequence used by Swarts and Jinek in crystallization, so the scissile phosphodiesters indicated were determined for a different sequence (see *Appendix 2—figure 1*, *Appendix 2—figure 1—figure supplement 6*) and superimposed onto the structural model according to their distance from the PAM (in terms of number of nucleotides). The discontinuity modeled into the non-target strand corresponds to positions of weak electron density in the crystal structure, which could have been due to some combination of disorder of the (intact) intervening tract and/or *in crystallo* hydrolysis and dissociation of the intervening tract. (**B**) For Cas12a, successful R-loop formation results in activation of the RuvC DNase active site to cleave three classes of DNA substrates (yellow scissors): the non-target strand (in *cis*), the target strand (in *cis*), and non-specific ssDNA (in *trans*). Circled numbers indicate the required order of *cis* strand cleavage; three conserved active site carboxylates of the RuvC DNase are shown in yellow and red; 'PAM' indicates the protospacer-adjacent motif; red arrow indicates the direction in which the R-loop is opened. Cas9 contains two DNase domains: the RuvC domain cleaves the non-target strand, and the HNH domain cleaves the target strand.

activated by specific target binding (*Figure 1B*; *Zetsche et al., 2015*; *Swarts et al., 2017*; *Chen et al., 2018*). Additionally, the position of the 20-nucleotide spacer sequence in Cas12a crRNAs is opposite to that in Cas9 crRNAs (3′ end versus 5′ end, respectively) (*Zetsche et al., 2015*; *Deltcheva et al., 2011*), and consequently, R-loop formation by each enzyme family occurs with opposing directionality (*Figure 1B*).

Cas12a belongs to the type V family of CRISPR effector proteins (whose names each begin with 'Cas12'), a classification defined by the presence of a single RuvC DNase domain (*Koonin et al., 2017*). Notably, although this classification is not strictly phylogenetic—the 'family' is actually a paraphyletic assembly of several evolutionarily independent protein lineages (*Koonin et al., 2017*; *Makarova et al., 2020*)—a second shared feature of all Cas12 interference complexes is the previously mentioned spacer-3′ crRNA architecture. However, the reason for this evolutionary convergence is unknown. Furthermore, while structures of DNA-bound Cas12a are known (*Yamano et al., 2016*; *Gao et al., 2016*; *Swarts et al., 2017*; *Stella et al., 2017*; *Stella et al., 2018*; *Swarts and Jinek, 2019*) and an obligatory order of strand cleavage has been suggested biochemically (*Swarts and Jinek, 2019*), the mechanism by which type V CRISPR enzymes form double-strand breaks using one active site remains unclear.

The RuvC domain of Cas12a exhibits stringent specificity for single-stranded substrates when activated for cleavage of free DNA in *trans* (Appendix 1) (*Chen et al., 2018*), suggesting that substrates cleaved in *cis* (i.e., the two DNA strands of a protein-bound R-loop) are also single-stranded during each of their respective cleavage events. Consistent with this substrate preference, in the most complete crystal structures of the Cas12a-bound R-loop, displacement of the non-target strand (NTS) of the DNA allows its association with the RuvC active site as a single strand (*Swarts et al., 2017*; *Swarts and Jinek, 2019*). This conformation likely permits initial non-target-strand cleavage, which is a prerequisite of DNA target-strand (TS) cutting (*Figure 1B*; *Swarts and Jinek, 2019*).

However, in these same crystal structures, the target-strand cleavage site is located within an ordered DNA duplex outside the R-loop,~25 Å away from the RuvC active site and inverted with respect to the most probable catalytic orientation (*Figure 1A*; *Swarts et al., 2017*; *Swarts and Jinek, 2019*). To satisfy these geometric constraints and the RuvC substrate preference, the target strand likely needs to separate from the non-target strand and bend, accessing a conformation evoked by some structures of the type V CRISPR-Cas enzymes Cas12b (C2c1) and Cas12e (CasX) (*Yang et al., 2016*; *Liu et al., 2019*). It has been hypothesized that contortion of the DNA substrate enables Cas12a (and other Cas12 family members) to cleave the target strand and complete its double-strand break (*Jeon et al., 2018*; *Zhang et al., 2019*), but it is unknown what enables this contortion.

We show here that Cas12a cleaves the target strand within a tract of DNA that is destabilized by the adjacent R-loop. Using chemical and fluorescent probes to investigate DNA conformation in protein-bound and protein-free R-loops in solution, we find that DNA flanking the RNA-3′ side of the R-loop exhibits signatures of single-strandedness, despite this region's potential for complete base pairing. The location of this DNA distortion controls the location of RuvC-catalyzed target-strand cleavage, suggesting that Cas12a exploits local duplex instability to complete its double-strand break. This cleavage mechanism is likely shared by other DNA-targeting CRISPR-Cas12 systems, which all use a single RuvC active site to cut the target strand within the DNA tract flanking the RNA-3′ side of the R-loop (*Yang et al., 2016*; *Liu et al., 2019*; *Yan et al., 2019*; *Karvelis et al., 2020*). Intriguingly, we find that nucleotides flanking the RNA-5′ side of protein-free R-loops remain

stably paired and stacked in solution. This fundamental asymmetry in nucleic acid structure offers a functional explanation for the puzzling convergence of type V CRISPR-Cas systems on the 5′-repeat-spacer-3′ crRNA orientation.

## Results and discussion

### Cas12a binding to DNA distorts the target-strand cleavage site

We suspected that dsDNA substrates of Cas12a would need to access a bent conformation to undergo target-strand cleavage. To chemically probe the structure of a Cas12a substrate in solution, we performed DNA permanganate footprinting on interference complexes containing a RuvC-inactivated mutant of a Cas12a ortholog from *Acidaminococcus species* (AsCas12a), hereafter called dCas12a. In this assay, permanganate selectively oxidizes thymines in non-B-form (e.g., locally melted or otherwise distorted) DNA structures, and oxidized positions are subsequently identified through piperidine-catalyzed strand cleavage (which occurs specifically at thymidine glycols) and denaturing polyacrylamide gel electrophoresis (PAGE) (*Figure 2A*; *Bui et al., 2003*). To enable sensitive detection of DNA fragments, we radiolabeled either the 5′ or 3′ end of each DNA strand (3′-end radiolabeling of DNA, which is not a common procedure, was achieved using a protocol developed for the present work, *Figure 2—figure supplement 1*). Consistent with previous applications of the permanganate assay to CRISPR-Cas-generated R-loops (*Xiao et al., 2017*), thymines within the portion of the non-target strand displaced by the crRNA were heavily oxidized, reflecting the single-strandedness of this DNA tract (*Figure 2A*).

Interestingly, we also observed significant oxidation at a thymine near the target-strand cleavage site (*Figure 2A*). To probe the region around the target-strand cleavage site more thoroughly, we adjusted the sequence of the DNA substrate to contain a stretch of A/T base pairs in the tract immediately adjacent to the R-loop, which we denote the R-loop flank (*Figure 2B*, *Figure 2—figure supplement 2*). We assessed the permanganate reactivity of the R-loop flank in three states of the Cas12a cleavage pathway: prior to Cas12a binding (apo DNA), after R-loop formation, and after the first set of cleavage events, which yield a 5-nt gap in the non-target strand (see Appendix 2 for details of the NTS gap). At each step, we observed an increase in permanganate reactivity on both strands of the DNA that persisted seven base pairs past the end of the crRNA, suggesting that Cas12a binding promotes distortion of DNA in the PAM-distal flank of the R-loop (*Figure 2B*, *Figure 2—figure supplement 3*, see Materials and methods).

In general, enhanced permanganate reactivity could reflect any of a variety of departures from B-form DNA duplex geometry, ranging from slight helical distortion to complete strand separation (*Bui et al., 2003*). As a result, the precise conformational ensemble of the R-loop flank cannot be absolutely determined from permanganate reactivity measurements. However, reactivity in the probed region was on the same order of magnitude as that of a fully single-stranded control, suggesting that the detected distortion involves substantial nucleobase unpairing and unstacking (*Figure 2B*, *Figure 2—figure supplement 3*, see Materials and methods). Additionally, permanganate reactivity decreased with distance from the R-loop edge (*Figure 2B*), consistent with NTS:TS fraying events that initiate from the R-loop edge (see Materials and methods). The increase in bulk permanganate reactivity in response to NTS cleavage may be due, at least in part, to increased binding occupancy of dCas12a/crRNA on the DNA substrate, as the NTS gap creates a high-energy interruption in the DNA rewinding pathway that boosts the stability of the ribonucleoprotein:DNA interaction (*Figure 2—figure supplement 2*; *Knott et al., 2019*). Thus, while the R-loop flank has the potential for complete base pairing, this DNA tract populates highly distorted, and probably unpaired, conformations when bound to Cas12a in solution.

### Distortion of the R-loop flank facilitates target-strand cleavage

DNA discontinuities as small as a nick or a single unpaired nucleotide are known to produce a 'swivel-like' point of flexibility within a duplex structure (*Mills et al., 1994*; *Le Cam et al., 1994*). We wondered whether the DNA distortion detected in the Cas12a R-loop flank may permit the global substrate bending and repositioning that is likely required for target-strand cleavage. A prediction of this model is that a change in the location of duplex instability should also change which phosphodiester bonds in the target strand are cleaved by the RuvC DNase. To test this prediction, we

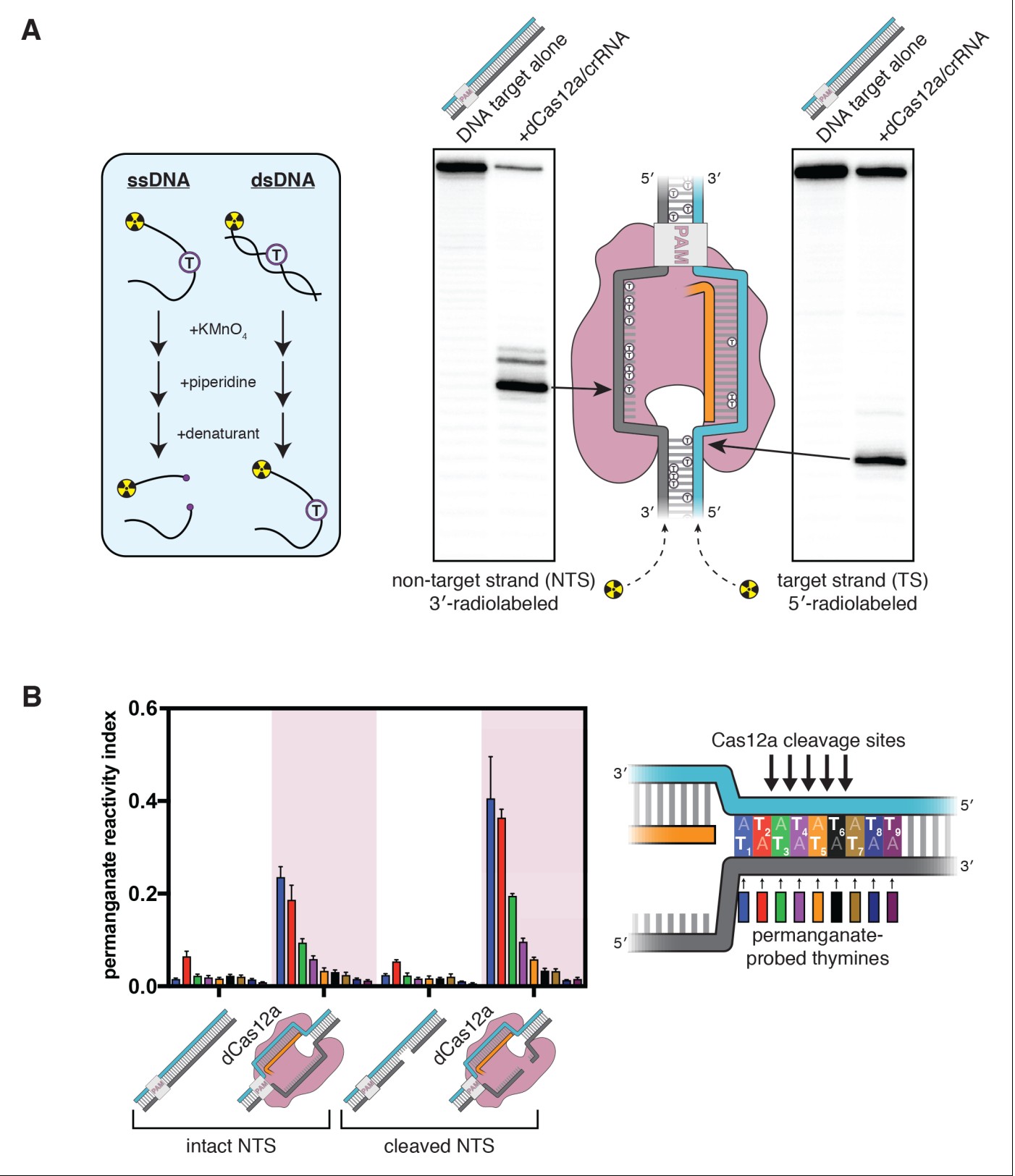

**Figure 2.** The target-strand cleavage site becomes distorted upon R-loop formation. (**A**) Denaturing PAGE phosphorimages of piperidine-treated permanganate oxidation products, demonstrating the assay's ability to detect non-B-form DNA conformations within and adjacent to a dCas12a-generated R-loop. Permanganate reactions were quenched after 10 s at 30°C. Each thymine in the DNA substrate is shown as a circled T. (**B**)
*Figure 2 continued on next page*

*Figure 2 continued*

Permanganate reactivity of a PAM-distal R-loop flank whose sequence was changed (as compared to the native protospacer sequence that was probed in **A**) to contain more thymines, with an intact or cleaved non-target strand ('cleaved NTS' indicates that there is a 5-nt gap in the NTS—see Appendix 2). Permanganate reactions were quenched after 2 min at 30°C. A raw phosphorimage is shown in *Figure 2—figure supplement 3*. The permanganate reactivity index (PRI) is an estimate of the rate of oxidation at each thymine, normalized such that PRI = 1 for a fully single-stranded thymine (see Materials and methods). Columns and associated error bars indicate the mean and standard deviation of three replicates. The phosphodiester bonds normally cleaved by WT Cas12a are indicated with arrows on the substrate schematic for reference, but note that the complexes being probed with permanganate were formed with dCas12a.

The online version of this article includes the following source data and figure supplement(s) for figure 2:

**Source data 1.** Numerical data plotted in *Figure 2* and accompanying figure supplements.
**Figure supplement 1.** Method used to 3′-end radiolabel DNA oligonucleotides.
**Figure supplement 2.** A gap in the non-target strand increases the affinity of dCas12a for its DNA target.
**Figure supplement 3.** Translating raw phosphorimages into quantitative permanganate reactivity metrics.

investigated the effect of R-loop size on the target-strand cut-site distribution. We reasoned that if DNA strand separation occurs at the R-loop edge, the location of the R-loop edge must define which nucleotides become unpaired and, consequently, which bonds can be accessed by the RuvC active site.

We first assessed our ability to shift the site of duplex instability by measuring permanganate reactivity in a truncated R-loop. To create a truncated R-loop, we mutated the DNA base pairs at positions 19 and 20, allowing crRNA strand invasion to progress only to the 18th nucleotide of the target sequence (position numbers indicate distance from the PAM). Additionally, we used a pre-cleaved non-target strand to lock the substrate in the chemical state that exists immediately before target-strand cleavage (Appendix 2). Finally, we prevented NTS:TS base pairing in the PAM-proximal part of the DNA target sequence by mutating the non-target strand at positions 1–12 (*Figure 3A*). This modification eliminated the branch-migration-catalyzed DNA dissociation pathway, allowing for the formation of uniformly stable complexes irrespective of PAM-distal crRNA:TS mismatches that would otherwise modulate affinity (*Strohkendl et al., 2018*; *Figure 3—figure supplement 1*). As a result, variations in the bulk permanganate reactivity of these constructs reflect variations in DNA structure rather than differential Cas12a/crRNA binding occupancy.

By implementing the permanganate assay on these complexes, we found that the A/T tract was highly reactive in the full R-loop but had limited reactivity in the truncated R-loop, suggesting that the distorted region had migrated with the edge of the R-loop (*Figure 3A*). We also observed this effect for a second set of crRNA/DNA sequences with equivalent base-pairing topology, demonstrating that this result is not unique to the originally tested sequence (*Figure 3—figure supplement 2*). Positions 19 and 20 of the DNA substrates were G/C base pairs, so DNA conformation at these nucleotides could not be assessed by permanganate reactivity. Nonetheless, these results show that nucleotide unpairing near the target-strand cleavage site depends not only on Cas12a binding and stable R-loop formation, but also on the extent of crRNA strand invasion (i.e., the size of the R-loop). Thus, by altering R-loop size, we can manipulate which nucleotides become unpaired upon Cas12a binding.

To test the hypothesis that distortion in the R-loop flank is linked to RuvC-mediated target-strand cleavage, we assembled wild type (WT) Cas12a with R-loops of various sizes and determined the distribution of target-strand cut sites. For these experiments, we used DNA substrates with an intact non-target strand that was mismatched with respect to the target strand throughout the region of crRNA complementarity (*Figure 3B*; DNA constructs containing such a mismatched tract are indicated with an asterisk—for example, 1–18* indicates a DNA construct whose target-strand sequence matches the crRNA sequence at positions 1–18 and whose non-target-strand is mismatched with respect to the target strand at the same positions). Preventing NTS:TS base pairing in this region allowed observation of DNA cleavage in enzyme-substrate complexes that would otherwise be too unstable to yield detectable rates of catalysis (*Strohkendl et al., 2018*; *Figure 3—figure supplement 3*).

As the R-loop edge was shifted toward the PAM, the target-strand cut-site distribution shifted toward the new R-loop edge (*Figure 3B*, *Figure 3—figure supplement 4*). When compensatory mutations were made in the crRNA to restore the original R-loop size, the target-strand cut sites

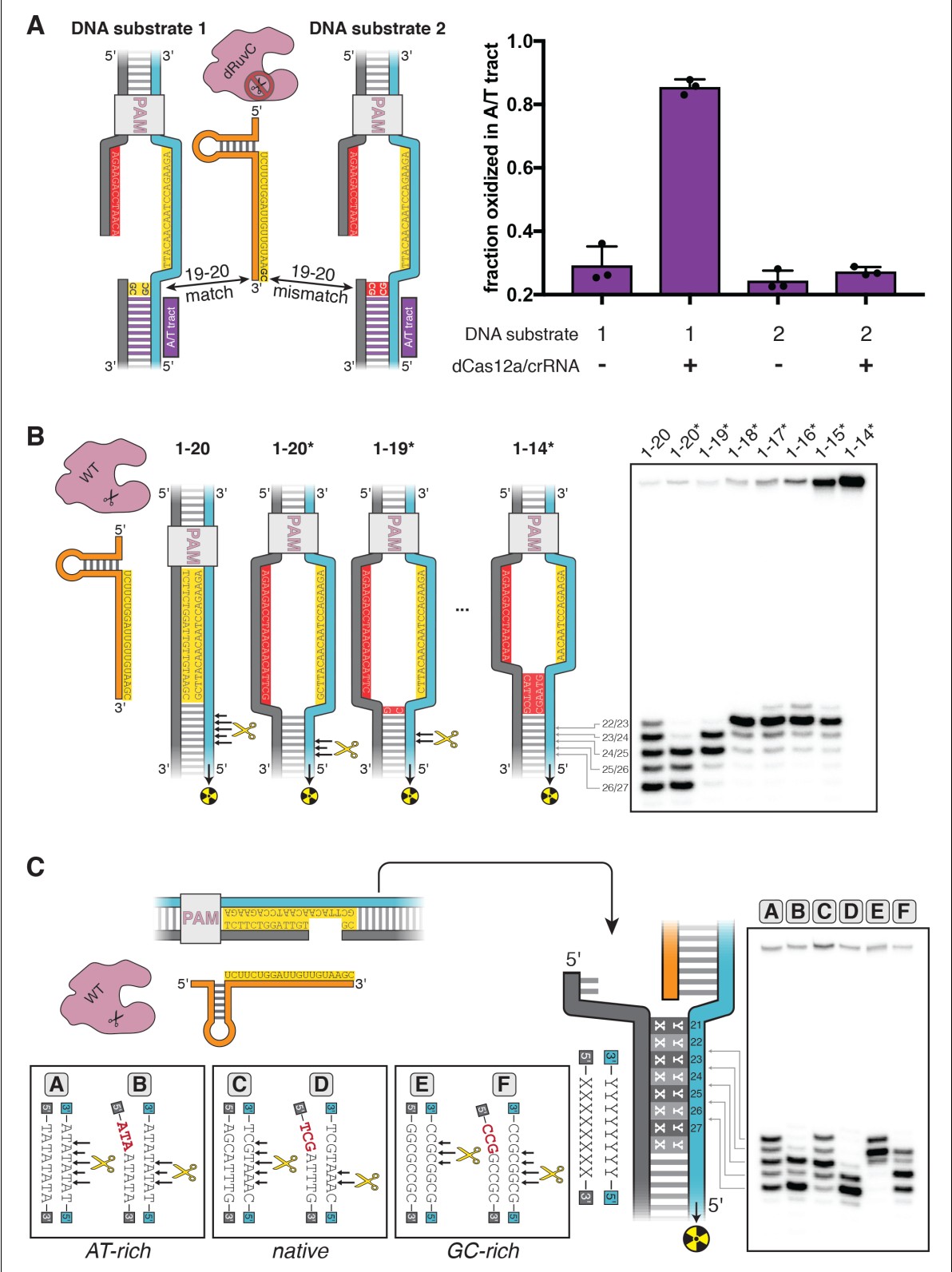

**Figure 3.** DNA distortion in the R-loop flank facilitates target-strand cleavage. (**A**) Permanganate reactivity of A/T tract in a 20-nt R-loop and an 18-nt R-loop. Permanganate experiments were conducted as in **Figure 2B** (2 minutes, 30°C). Purple rectangles alongside DNA schematics indicate the location of the tract of DNA whose permanganate reactivity is being quantified. The y-axis denotes the fraction of DNA molecules estimated to have been oxidized on at least one thymine within the A/T tract (see Materials and methods). Columns and associated error bars indicate the mean and

*Figure 3 continued on next page*

*Figure 3 continued*

standard deviation of three replicates. (B) Target-strand cut-site distribution with a shrinking R-loop, as resolved by denaturing PAGE and phosphorimaging (n = 3). 100 nM AsCas12a and 120 nM crRNA were incubated with 1 nM of DNA target at 37°C for 1 hr, prior to quenching and resolution by denaturing PAGE (kinetics shown in *Figure 3—figure supplement 4*). Each lane corresponds to a different DNA target, bearing varying numbers of PAM-distal mismatches with respect to the crRNA. Indicated above each lane is the number of base pairs of complementarity between the target strand and the crRNA spacer, starting with the base immediately adjacent to the PAM. For the lane lacking an asterisk, the DNA target was fully duplex. For the lanes that bear asterisks, the DNA target contained a bubble across the region of crRNA:TS complementarity, which stabilized the interaction of the DNA with the Cas12a/crRNA complex. Numbers to the left of the phosphorimage indicate the position (distance from the PAM, as numbered in C) of the dinucleotide whose phosphodiester was cleaved to yield the labeled band. Black arrows are drawn on the substrate diagrams to indicate cleaved phosphodiesters (as determined from the phosphorimage), and relative arrow lengths are roughly reflective of relative band intensities. (C) Target-strand cut-site distribution with various sequences in the R-loop flank (all with a 20-nt R-loop), as resolved by denaturing PAGE and phosphorimaging (n = 3). 100 nM AsCas12a and 120 nM crRNA were incubated with 1 nM of DNA target at 25°C for 10 min, prior to quenching and resolution by denaturing PAGE (kinetics shown in *Figure 3—figure supplement 7*). All DNA targets were 5'-radiolabeled on the target strand. The non-target strand contained a gap from positions 14–18 (see Appendix 2) but was complementary to the target strand at positions 1–13 and 19–20. In each lane, the DNA target was varied to contain different sequences in the R-loop flank, which either formed a perfect duplex (substrates A, C, and E) or contained a 3-bp NTS:TS mismatch (substrates B, D, and F). Black arrows are drawn on the substrate diagrams as in B.

The online version of this article includes the following source data and figure supplement(s) for figure 3:

**Source data 1.** Numerical data plotted in *Figure 3* and accompanying figure supplements.
**Figure supplement 1.** dCas12a ribonucleoprotein binds tightly to pre-gapped/pre-unwound targets despite PAM-distal mismatches.
**Figure supplement 2.** Effect of R-loop truncation on permanganate reactivity of the A/T tract.
**Figure supplement 3.** Effect of PAM-distal mismatches on non-target-strand and target-strand cleavage kinetics and position with fully duplex DNA targets.
**Figure supplement 4.** Effect of PAM-distal mismatches on non-target-strand and target-strand cleavage kinetics and position with bubbled DNA targets.
**Figure supplement 5.** Determinants of altered target-strand cleavage kinetics and position.
**Figure supplement 6.** Non-target-strand cut-site distribution with a shrinking R-loop.
**Figure supplement 7.** Kinetics of target-strand cleavage in DNA targets with various sequences in the R-loop flank.

moved back toward their original distribution (*Figure 3—figure supplement 5*). The observed shifts in the cut sites were not due to general destabilization of the R-loop, as a single crRNA:TS mismatch at an internal position of the target sequence (position 9) slowed cleavage without affecting cut-site distribution (*Figure 3—figure supplement 5*). The cut-site distribution shared by the 1–17*, 1–16*, and 1–15* substrates, along with the lack of cleavage of the 1–14* substrate, may reveal a geometric limit on bent DNA conformations that still permit active site association (*Figure 3B*). Additionally, the broader target-strand cut-site distribution in the DNA target lacking NTS:TS mismatches (labeled '1–20' in *Figure 3B*) could reflect bending events that initiated from partially rewound R-loop conformations. Notably, the non-target-strand cut-site distribution did not change markedly as the R-loop was truncated, suggesting that non-target-strand cleavage is unrelated to nucleotide unpairing in the R-loop flank (*Figure 3—figure supplement 6*). These results imply that the site of Cas12a-mediated target-strand cleavage is tied to, and perhaps dictated by, the location of weakened base pairing. Consistent with this idea, a previous study found a linkage between Cas12a cut-site distribution and nucleic acid conformation as measured by Förster resonance energy transfer (*Zhang et al., 2019*).

These principles predict that R-loop flank sequences with greater nucleobase stacking energy should limit the depth of fraying and, consequently, favor target-strand cleavage events that are closer to the PAM. In agreement with this prediction, of three DNA targets that differed only in the sequence of their R-loop flank—native protospacer, AT-rich, or GC-rich—the GC-rich substrate was cleaved most PAM-proximally (*Figure 3C*). Additionally, eliminating NTS:TS base pairing at positions 21–23 led to fast and PAM-distally shifted cleavage of the target strand in all cases (*Figure 3C*, *Figure 3—figure supplement 7*). Together, these results suggest that DNA distortion in the R-loop flank is an important enabler of Cas12a-catalyzed target-strand cleavage.

## Duplex instability is intrinsic to DNA in the RNA-3' flank of R-loops

Next, we wondered what role the Cas12a protein plays in distortion of the R-loop flank. To assess the contribution of the protein, we formed a protein-free mimic of the nucleic acid structure immediately prior to target-strand cleavage. This artificial R-loop contained a pre-cleaved non-target strand

that was mismatched with respect to the target strand in the 20-nt stretch adjacent to the PAM, and the same stretch of the target strand was hybridized to a complementary 20-nt RNA oligonucleotide (*Figure 4A*). When we probed the permanganate reactivity of this protein-free R-loop, we found that the A/T tract was slightly more reactive than in the Cas12a-generated R-loop experiment (*Figure 4A*, *Figure 4—figure supplement 1*).

To interpret the reactivity of an R-loop mimic, we used the RNA-free DNA bubble control as a point of comparison. This bubbled DNA substrate reveals that permanganate reactions occur readily at the edge of the bubble, that is, the terminus of the DNA homoduplex (substrate C in *Figure 4—figure supplement 1*, also see Materials and methods). We therefore asked whether the adjacent RNA:DNA hybrid protects or sensitizes the DNA:DNA terminus as compared to the RNA-free control. The naïve prediction is that a DNA duplex terminus should become more stable (i.e., less prone to end fraying) when another duplex is stacked on top of it (*Häse and Zacharias, 2016*). It is thus surprising that the RNA oligonucleotide in this experiment sensitizes the DNA:DNA terminus to oxidation by permanganate. This result indicates that distortion of the R-loop flank is a phenomenon intrinsic to the R-loop boundary itself and that the protein only needs to hold the R-loop open to promote flexibility in the adjacent DNA.

## Duplex instability is not a feature of DNA in the RNA-5′ flank of R-loops

While Cas9 also conducts R-loop-dependent DNA cleavage, its R-loop topology is inverted with respect to that of Cas12a as a result of their opposing crRNA architectures—Cas12a crRNAs occur as 5′-repeat-spacer-3′, whereas Cas9 crRNAs occur as 5′-spacer-repeat-3′ (*Figure 1B*). Given the instability of the Cas12a R-loop flank (referred to as a 3′ R-loop flank because it contains a 3′ RNA terminus), we wondered whether the PAM-distal flank of the Cas9 R-loop (a 5′ R-loop flank) would also be unstable.

To test this question, we assayed flank distortion in an R-loop created by a catalytically inactive mutant of Cas9 from *Streptococcus pyogenes* (dCas9) and in the corresponding protein-free mimic (the non-target strand was pre-cleaved analogously to a Cas12a substrate). Remarkably, we found that the flank experienced nearly background oxidation levels both in the protein-bound R-loop (with dCas9 at a saturating concentration, *Figure 4—figure supplement 2*) and in the protein-free mimic, suggesting that unlike 3′ R-loop flanks, 5′ R-loop flanks are not naturally unstable (*Figure 4A*, *Figure 4—figure supplement 1*). The 5′ R-loop flank behaved consistently with expectations about coaxial duplex stacking, as the RNA oligonucleotide protected the DNA:DNA duplex terminus as compared to the bubbled control (*Figure 4—figure supplement 1*). Thus, an RNA:DNA hybrid can either stabilize or destabilize a juxtaposed DNA:DNA duplex terminus, depending on whether the hybrid terminus contains a 5′ RNA end or a 3′ RNA end, respectively. These results suggest a fundamental energetic difference in the conformational landscapes of 3′ versus 5′ R-loop flanks (*Figure 4B*).

The conformational difference between 3′ and 5′ R-loop flanks is intrinsic to strand polarity, as the trends in permanganate reactivity were robust to changes in nucleic acid sequence, end chemistry, and non-target-strand cleavage state (*Figure 4—figure supplements 3–6*). Additionally, we detected the same polarity dependence when we measured fluorescence intensity of a single 2-aminopurine nucleotide present at position 21 of the original protospacer sequence, indicating that the conformational difference is not an artifact of the permanganate reactivity assay or of the AT-rich sequence of the modified protospacer (*Figure 4—figure supplement 7*). Therefore, while Cas12a does not seem to actively destabilize the R-loop flank, the protein forms R-loops with the topology that natively yields a greater degree of flexibility in the region beyond the end of the crRNA.

## Differences in interhelical stacking energy may underlie asymmetric R-loop flank stability

Seeking a mechanistic explanation for the unequal stability of 3′ versus 5′ R-loop flanks, we hypothesized that the asymmetry may emerge from energetic differences in the coaxial stacking of a DNA homoduplex on either end of an RNA:DNA hybrid (*Figure 5*). Because interhelical junctions are known to explore both stacked and unstacked conformations in solution (*Roll et al., 1998*; *Protozanova et al., 2004*), they are expected to populate each state to a degree that depends on the free energy change associated with coaxial duplex stacking. Weaker stacking energy, then,

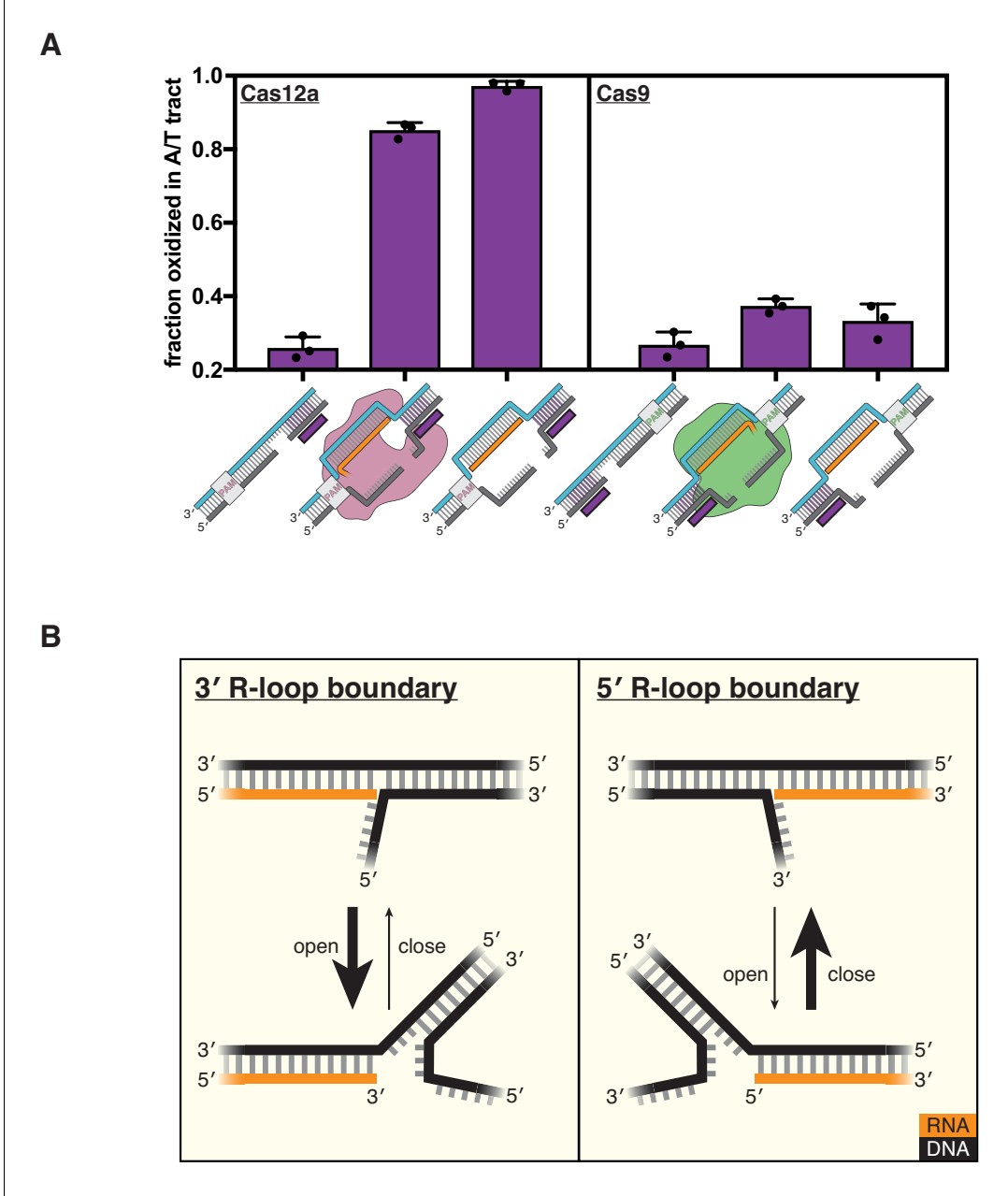

**Figure 4.** DNA distortion is protein-independent and unique to 3' R-loop flanks. (**A**) Permanganate reactivity of the A/T tract in a dCas12a R-loop, a dCas9 R-loop, and their protein-free mimics. The y-axis denotes the fraction of DNA molecules estimated to have been oxidized on at least one thymine within the A/T tract (see Materials and methods). Purple rectangles alongside DNA schematics indicate the location of the tract of DNA whose permanganate reactivity is being quantified. Columns and associated error bars indicate the mean and standard deviation of three replicates. (**B**) Model for the relative conformational dynamics of 3' and 5' R-loop boundaries, as suggested by permanganate reactivity experiments. The depth of fraying shown (three base pairs) was chosen arbitrarily for the schematic and should not be interpreted as a uniquely stable 'open' structure (see Materials and methods).

The online version of this article includes the following source data and figure supplement(s) for figure 4:

**Source data 1.** Numerical data plotted in **Figure 4** and accompanying figure supplements.
**Figure supplement 1.** Permanganate reactivity of the A/T tract in R-loops formed by dCas12a or dCas9.
**Figure supplement 2.** dCas9 binds tightly to pre-gapped DNA targets.
**Figure supplement 3.** Permanganate reactivity of the A/T tract in protein-free R-loops of various sequences.
**Figure supplement 4.** Effect of RNA end chemistry on permanganate reactivity of the A/T tract in protein-free R-loops.
**Figure supplement 5.** Asymmetry in R-loop flank stability is also a feature of intact R-loops.
**Figure supplement 6.** Effect of overhanging non-target-strand nucleotides on permanganate reactivity of the A/T tract in protein-free R-loops.
*Figure 4 continued on next page*

*Figure 4 continued*

**Figure supplement 7.** 2-aminopurine fluorescence measurements confirm asymmetry in conformational dynamics of R-loop flanks.

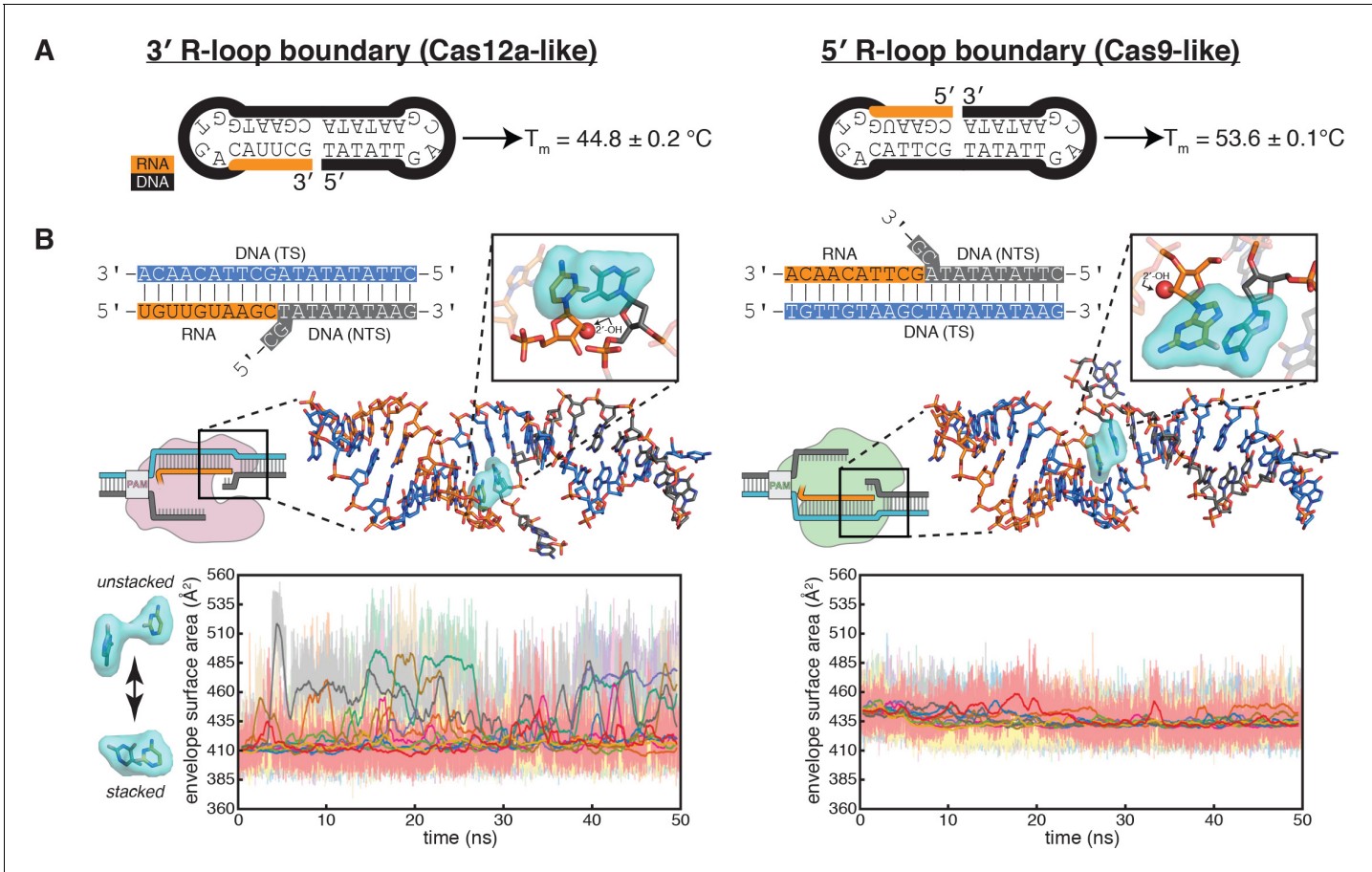

**Figure 5.** Energetics of base stacking at the R-loop boundary probed by optical measurements and molecular dynamics simulations. (**A**) Melting temperatures of nicked-dumbbell constructs that recapitulate each type of R-loop boundary, determined by monitoring absorbance of ultraviolet light while slowly cooling samples from 95°C to 2°C. Reported values show mean and standard deviation of three replicates. See *Figure 5—figure supplement 1* for refolding curves and control constructs. (**B**) Molecular dynamics simulations reveal nucleobase unstacking in 3' R-loop boundaries but not in 5' R-loop boundaries. At the top left is a schematized version of the true structural model shown immediately below (this coaxially stacked conformation is the starting structure that was used for simulation); hydrogens were present in the simulated model and analyses but are omitted from representations here for clarity. The simulated model contained only the nucleic acid molecules shown in stick representation; the protein and remainder of the R-loop are drawn in a schematic only to orient the reader as to where the simulated structure would fit into a full DNA-bound CRISPR interference complex; the Cas9-orientation R-loop is drawn with a Cas12a-like NTS gap to reflect the simulated model. The inset is a closeup of the two nucleotides on the 'flapped' side of the junction in the structural model; the 2'-OH is shown as a red sphere. Envelope surface area (ESA) was determined by isolating two nucleobases of the interhelical stack—that on the RNA terminus and that stacked upon it from the NTS—and calculating the surface area of the volume they jointly occupy over the course of each trajectory (envelope shown in cyan). High ESA values reflect unstacking of nucleobases, whereas low ESA values reflect a stacked architecture similar to that of the starting conformation. Pale lines are absolute ESA values, and bold lines are moving averages (1-ns sliding window). Data from ten independent 50-ns trajectories are shown in different colors. Simulations of a second set of sequences are described in *Figure 5—figure supplement 2*.

The online version of this article includes the following source data and figure supplement(s) for figure 5:

**Source data 1.** Numerical data plotted in *Figure 5* and accompanying figure supplements.
**Source data 2.** Example molecular dynamics trajectories.
**Figure supplement 1.** Thermal stability determination for nicked dumbbell substrates and their constituent hairpins.
**Figure supplement 2.** Molecular dynamics simulations of the Cas12a-like and Cas9-like interhelical junctions, Sequence 2.

encourages exploration of motions that initiate preferentially from unstacked duplex termini, such as DNA bending (*Protozanova et al., 2004*) and fraying (*Häse and Zacharias, 2016*).

To investigate whether differences in interhelical stacking energy could explain the difference in flank stability of the two R-loop topologies, we designed dumbbell substrates that reduced each type of R-loop boundary to a single chimeric oligonucleotide that contains both an RNA:DNA hybrid and a DNA:DNA homoduplex (*Figure 5A*, *Figure 5—figure supplement 1*). A stronger interhelical stack in these dumbbells should increase the thermal stability of the folded state (*Erie et al., 1987*). Through temperature-dependent hyperchromicity measurements, we determined that the RNA-5′ dumbbell (resembling the PAM-distal R-loop edge of Cas9) had a melting temperature 9°C higher than that of the RNA-3′ dumbbell (resembling the PAM-distal R-loop edge of Cas12a) (*Figure 5A*, *Figure 5—figure supplement 1*). The observed difference in melting temperature supports the idea that the resistance of 5′ R-loop flanks to permanganate oxidation may emerge from a more stable interhelical stack.

To probe the structural and energetic features of the interhelical stacks in atomic detail, we built models of coaxially stacked interhelical junctions of the two types: one containing an RNA-3′ end and one containing an RNA-5′ end. We performed a total of 500 nanoseconds of molecular dynamics simulation on each model, and we performed a second set of simulations on models of a different nucleotide sequence. Strikingly, the 3′ R-loop junctions frequently unstacked over the course of these short simulations, while the 5′ R-loop junctions remained relatively stable (*Figure 5B*, *Figure 5—figure supplement 2*). Observation of large-scale conformational transitions relevant to Cas12a-mediated DNA cleavage, such as fraying or bending events, would likely require much longer simulations (*Leroy et al., 1988*), but these short simulations suggest that the experimentally observed instability of 3′ R-loop flanks may arise from frequent unstacking events that comprise an early step in the DNA bending pathway. We speculate that different levels of interhelical stacking energy may emerge from certain sequence-independent features of the two duplexes juxtaposed in each type of R-loop boundary, such as helical geometry. The difference in the stacking equilibrium then leads to an unequal propensity for base pairing in the flanking DNA homoduplex. However, whether a difference in interhelical stacking energy can fully explain the observed difference in fraying propensity will require more detailed experimentation and analysis.

## Concluding remarks

Our analyses have uncovered a fundamental asymmetry in the structure of the two kinds of R-loop flanks—RNA-3′ flanks exhibit a distorted conformation, whereas RNA-5′ flanks resemble standard B-form DNA (*Figure 4B*)—that is directly relevant to the mechanism of Cas12a-mediated DNA double-strand break formation. Specifically, the ability of the Cas12a RuvC active site to capture the target strand may rely on the active site's proximity to the uniquely malleable 3′ R-loop flank, which relies in turn upon the topology of the Cas12a R-loop (*Figure 6*).

This peculiar asymmetry in nucleic acid conformational dynamics may also have consequences for non-CRISPR-associated R-loops. R-loops are common byproducts of transcription in eukaryotic genomes, and their dysregulation has been linked to a number of diseases (*Crossley et al., 2019*). It has been proposed that the mode of pathogenesis, in some cases, may emerge from the aberrant activity of general DNA repair nucleases on R-loop boundaries, whose scission leads to the formation of toxic double-strand breaks (*Sollier et al., 2014*). Notably, in the earliest biochemical exploration of this process, purified XPG (a human DNA repair nuclease) and nuclease P1 (a secreted fungal nuclease commonly used to selectively degrade single-stranded nucleic acid structures) robustly cleaved a 3′ R-loop flank but left the 5′ R-loop flank largely untouched (*Tian and Alt, 2000*). While this result was originally attributed to an idiosyncrasy of the tested sequence, our results suggest that the flank preference may have been polarity-dependent, not sequence-dependent. The experiments of Tian and Alt could represent an independent observation of the phenomenon dissected in the present work, implying that the asymmetric structure of R-loop boundaries may affect DNase sensitivity and genome stability in multiple domains of life.

Cas12a's mode of dsDNA targeting contrasts with the established DNA cleavage mechanisms of other CRISPR interference complexes, which do not rely on instability in DNA flanking the R-loop. Cas9 forms R-loops with the topology that yields a stably base-paired PAM-distal R-loop flank (*Figure 4A*), but its second nuclease domain obviates the need to cleave outside the R-loop (*Figure 1B*). In type I CRISPR interference complexes, which have the same R-loop topology as

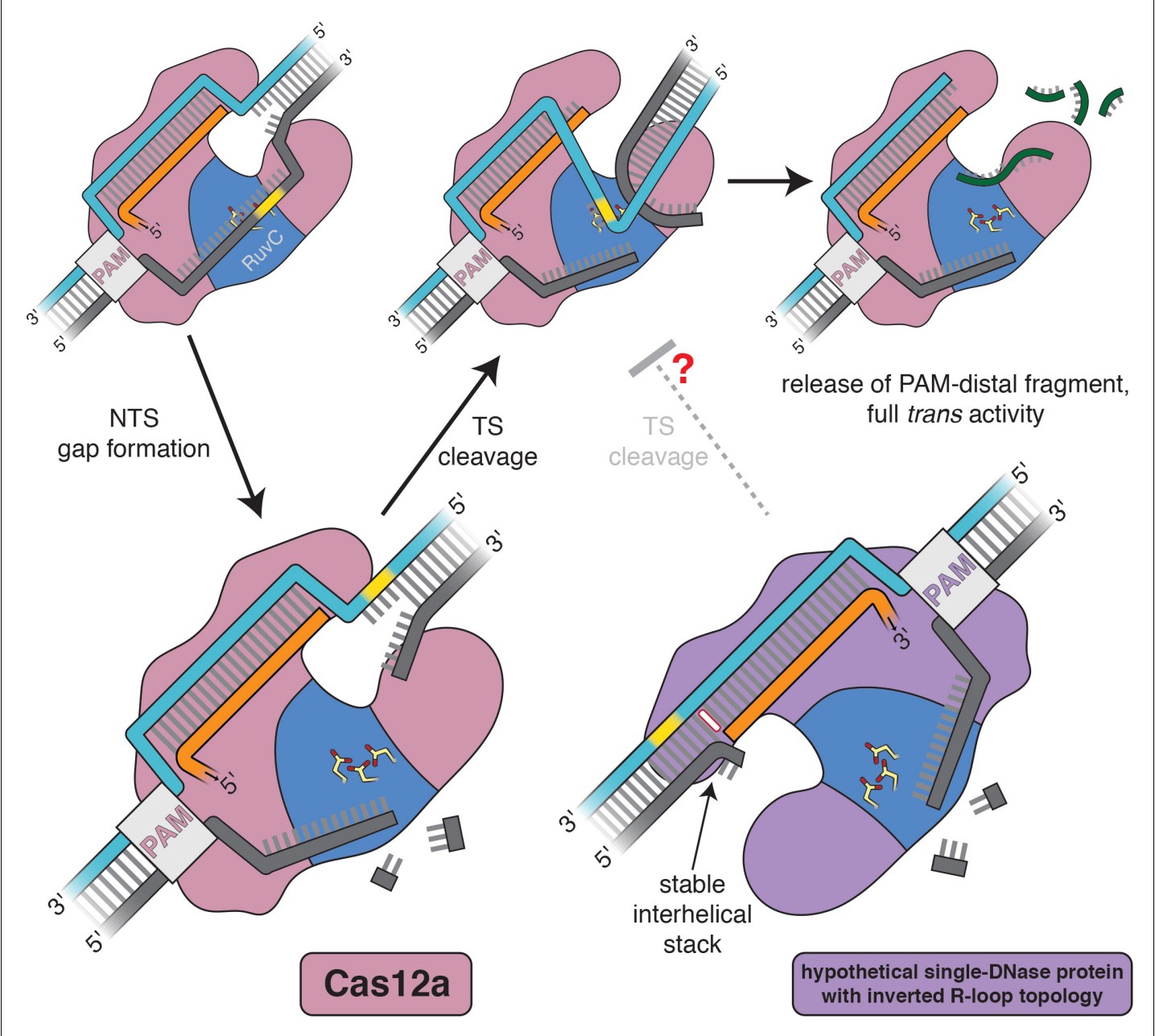

**Figure 6.** Model for the double-strand-break formation pathway of Cas12a and that of an analogous (hypothetical) enzyme with inverted R-loop topology. Scissile DNA tracts are shown in yellow. The stable interhelical stack in the hypothetical inverted complex is highlighted in white.

Cas12a, the single-strand-specific DNase Cas3 is used to nick the displaced portion of the non-target strand after R-loop formation (*Westra et al., 2012*), similar to the initial non-target-strand nicking event in Cas12a. Cas3 eventually gains access to the PAM-proximal R-loop flank for processive DNA degradation, but, importantly, it uses an ATP-driven helicase to do so (*Mulepati and Bailey, 2013*). Thus, DNA cleavage in the 3′ R-loop flank of the Cas12a interference complex seems to be a solution that maximizes the utility of its minimal enzymatic machinery.

Intriguingly, all known DNA-targeting Cas12 enzymes, which are taxonomically defined by their minimal enzymatic machinery (a single RuvC DNase domain), form R-loops of the topology that allows target-strand cleavage within a PAM-distal 3′ R-loop flank (*Yang et al., 2016*; *Liu et al., 2019*; *Yan et al., 2019*; *Karvelis et al., 2020*) (for target-strand cleavage properties of Cas12i, see

Appendix 2). Given the importance of DNA bending in enabling a single RuvC active site to form double-strand breaks, we propose that the universal type V R-loop topology may be the product of biophysically imposed selective pressures. If a single-DNase Cas enzyme were to form R-loops of the Cas9-like topology, the target strand would be cut, by analogy, within a 5′ R-loop flank. Our results indicate that 5′ R-loop flanks resist fraying and bending, which may lead to slower target-strand cleavage in the hypothetical topology-inverted enzyme (*Figure 6*). We hypothesize that, under the strong evolutionary pressure for immune responses that efficiently produce double-strand breaks (*Heitman et al., 1999*), ancestral enzymes with PAM-distal 3′ R-loop flanks outperformed topology-inverted variants, perhaps leading to the modern ubiquity of the Cas12a-like R-loop architecture in type V enzymes. Importantly, the extant diversity of Cas12 proteins likely did not emerge from a single CRISPR-associated ancestor but rather from many distinct transposon-encoded TnpB proteins (*Koonin et al., 2017*; *Makarova et al., 2020*), suggesting that several independent evolutionary trajectories have converged on the same R-loop topology.

While these evolutionary speculations cannot be experimentally verified at present, our findings also provide valuable mechanistic information that will support the development of Cas12-based genome manipulation technologies. For instance, it has been proposed that Cas12-family proteins could be used as RNA-guided DNA 'nickases' through selective removal of target-strand cleavage activity, but those nickases reported so far are catalytically inefficient and do not totally eliminate target-strand cutting (Appendix 2) (*Yamano et al., 2016*; *Yan et al., 2019*). Our permanganate reactivity results revealed that target-strand distortion in Cas12a-generated R-loops, as probed by our experimental techniques, can be explained by conformational dynamics intrinsic to the nucleic acids (*Figure 4A*). Thus, engineering a type V CRISPR nickase that performs fast non-target-strand cleavage but undetectable target-strand cleavage will likely require modifications that shield or distance the RuvC active site from the intrinsically labile 3′ R-loop flank. Looking further ahead, if *de novo* design of Cas-like RNA-guided nucleases (*Rauch et al., 2019*) someday enters the realm of dsDNA-targeting, knowledge of the asymmetry in R-loop flank stability may be useful in tailoring the architecture of a designed enzyme to a specific application. Our results suggest that placing a DNase domain near a 3′ R-loop flank would encourage fast target-strand cleavage (for applications requiring double-strand breaks), while placing one near a 5′ R-loop flank would likely inhibit target-strand cleavage (for nickase applications). Complex engineering feats will become more attainable through continued investigation into the functional, structural, and evolutionary features of natural dsDNA-targeting CRISPR systems.

# Materials and methods

### Key resources table

| Reagent type (species) or resource | Designation | Source or reference | Identifiers | Additional information |
|---|---|---|---|---|
| Recombinant DNA reagent | pMBP-AsCas12a expression plasmid | *Chen et al. (2018)* | RRID: Addgene_113430 | See *Supplementary file 1* for the sequences of all plasmids and oligonucleotides |
| Peptide, recombinant protein | T4 polynucleotide kinase | New England Biolabs | M0201S | |
| Peptide, recombinant protein | T4 RNA ligase 2 | New England Biolabs | M0239S | |
| Peptide, recombinant protein | AsCas12a protein | this paper | | All Cas proteins purified from *E. coli* BL21 Star(DE3) cells |
| Chemical compound, drug | [γ-$^{32}$P]-ATP | PerkinElmer | BLU502A001MC | |
| Chemical compound, drug | potassium permanganate | Sigma-Aldrich | 223468 | |
| Software, algorithm | ImageQuant TL | GE Healthcare | RRID:SCR_014246 | |

## Protein expression and purification

Expression plasmids were cloned as previously described (*Chen et al., 2018*; *Knott et al., 2019*). Briefly, protein-coding DNA segments were shuttled into custom pET-based vectors by Gibson assembly, and catalytic mutants were generated through site-directed mutagenesis by polymerase chain reaction (PCR) with blunt-end ligation (plasmid sequences are in *Supplementary file 1*). The parent plasmid from which the Cas12i1 expression plasmid was cloned (pET28a-mH6-Cas12i1) was a gift from Arbor Biotechnologies (Addgene plasmid #120882). AsCas12a protein expression and purification was performed as previously described (*Chen et al., 2018*) with the following modifications. The cells used for expression were *E. coli* BL21 Star(DE3). Lysis buffer was 50 mM HEPES (pH 7.5), 500 mM NaCl, 1 mM TCEP, 0.5 mM PMSF, 10 tablets/L cOmplete EDTA-free protease inhibitor cocktail (Roche), and 0.25 mg/mL chicken egg white lysozyme (Sigma-Aldrich). Ni-NTA wash buffer was 50 mM HEPES (pH 7.5), 500 mM NaCl, 1 mM TCEP, 5% glycerol, 20 mM imidazole. Ni-NTA elution buffer was 50 mM HEPES (pH 7.5), 500 mM NaCl, 1 mM TCEP, 5% glycerol, 300 mM imidazole. TEV protease cleavage was performed overnight while dialyzing against dialysis buffer (50 mM HEPES (pH 7.5), 250 mM NaCl, 1 mM TCEP, 5% glycerol). Low-salt ion exchange buffer was 50 mM HEPES (pH 7.5), 250 mM KCl, 1 mM TCEP, 5% glycerol. High-salt ion exchange buffer was 50 mM HEPES (pH 7.5), 1 M KCl, 1 mM TCEP, 5% glycerol. Gel filtration buffer was 20 mM HEPES (pH 7.5), 200 mM KCl, 1 mM TCEP, 5% glycerol. Cas12i1 was expressed and purified with the same protocol as for AsCas12a. FnCas12a and SpCas9 were expressed with the same protocol as for AsCas12a. Purification of FnCas12a only differed from that of AsCas12a in the lack of a TEV protease cleavage step, as the FnCas12a expression construct lacked a cleavable tag. Purification of SpCas9 only differed from that of AsCas12a in the ion exchange buffers and gel filtration buffer, which contained 10% glycerol instead of 5% glycerol. Each protein was expressed and purified once to create a single set of stock aliquots that were used for all experiments reported within this manuscript.

## *In vitro* transcription of RNA

Guide RNAs and some short RNAs used in R-loop mimics were produced by *in vitro* transcription (see *Supplementary file 1*). Double-stranded DNA templates for T7 RNA polymerase transcription were assembled from several overlapping DNA oligonucleotides (synthesized by IDT) by PCR. Transcription occurred in 40 mM Tris-Cl (pH 7.9 at 25°C), 25 mM $MgCl_2$, 10 mM dithiothreitol, 0.01% (v/v) Triton X-100, and 2 mM spermidine, with 5 mM of each NTP and 100 μg/mL T7 RNA polymerase. Transcription was allowed to proceed for 2.5 hr at 37°C. Cas9 sgRNAs were transcribed with a hammerhead ribozyme on the 5′ end to allow an arbitrary sequence on the 5′ end of the final sgRNA. Cas12a and Cas12i crRNAs were transcribed with a hepatitis delta virus ribozyme on the 3′ end to avoid the impurities associated with non-templated nucleotide addition in the final crRNA. 20-nt spacers used in R-loop mimics were transcribed with both a 5′-HHrz and a 3′-HDVrz. For those RNA transcripts that contained hammerhead ribozyme, which is prone to misfolding, an additional 5 mM $MgCl_2$ was added to the transcription products, and the reaction was placed on a thermocycler for iterative annealing ({80°C for 2 min, 37°C for 10 min} x 8, hold at 10°C). DNA in all transcription reactions was then digested with DNase I (RQ1 RNase-Free DNase, Promega) (0.05 U/μL, manufacturer's units) for 30 min at 37°C. RNA fragments released by the ribozymes were then purified by denaturing PAGE (10% acrylamide:bis-acrylamide 29:1, 7 M urea, 0.5X TBE), ethanol-precipitated, and resuspended in RNA storage buffer (0.1 mM EDTA, 2 mM sodium citrate, pH 6.4).

## Nucleic acid and interference complex preparation

All DNA oligonucleotides and some RNA oligonucleotides (as indicated in *Supplementary file 1*) were ordered from Integrated DNA Technologies. DNA oligonucleotides used in biochemical experiments were PAGE-purified in house and resuspended in water. $A_{260}$ was measured on a Nano-Drop (Thermo Scientific), and concentration was estimated according to extinction coefficients determined by OligoCalc (*Kibbe, 2007*). DNA substrates were annealed in annealing buffer (10 mM Tris-Cl, pH 7.9 at 25°C, 50 mM KCl, 1 mM EDTA) by heating to 95°C and cooling to 25°C over the course of 40 min on a thermocycler. This annealing reaction was always performed with 40 nM radiolabeled DNA strand. When the DNA substrate comprised just two complementary strands, the unlabeled (complementary) strand was included in the annealing reaction at 80 nM. When the DNA substrate comprised more than two distinct strands, strands with the same sense as the radiolabeled

strand were included at 80 nM, and strands complementary to the radiolabeled strand were included at 60 nM. For protein-free R-loop mimics, the RNA (or other spacer mimic) was included in the annealing reaction at 400 nM, to match the guide RNA concentration of a protein-containing experiment. The substrate concentrations reported in figure legends refer to the concentration of the radiolabeled strand. All crRNA and sgRNA molecules were annealed in RNA storage buffer (0.1 mM EDTA, 2 mM sodium citrate, pH 6.4) prior to use (80℃ for 1 min, then moved directly to ice). To form CRISPR surveillance complexes (Cas protein + guide RNA), crRNA or sgRNA was combined with Cas protein (both at 2X final concentration in 1X reaction buffer) and allowed to equilibrate for 5 min at 37℃. To form complete interference complexes (Cas protein + guide RNA + DNA target), 1 vol of 2X surveillance complex (in 1X reaction buffer) was combined with 1 vol of 2X DNA substrate (in 1X reaction buffer) and allowed to equilibrate for 5 min at 37℃ (if applicable).

## DNA oligonucleotide radiolabeling

Standard 5′ radiolabeling was performed with T4 polynucleotide kinase (New England Biolabs) at 0.2 U/µL (manufacturer's units), 1X T4 PNK buffer (New England Biolabs), 400 nM DNA oligonucleotide, and 200 nM [γ-$^{32}$P]-ATP (PerkinElmer) for 30 min at 37℃, followed by a 20-min heat-killing incubation at 65℃. Radiolabeled oligos were then buffer exchanged into water using a Microspin G-25 spin column (GE Healthcare). For 3′ radiolabeling (see *Figure 2—figure supplement 1*), which was based on the mechanistic work of *Nandakumar and Shuman (2004)*, the DNA oligonucleotide to be radiolabeled was synthesized by Integrated DNA Technologies with two modifications: the sugar moiety of the 3′-most nucleotide was a ribose, and the sugar moiety of the penultimate nucleotide was a 2′-O-methyl ribose. A 'phosphate shuttle' RNA oligonucleotide underwent a high-yield 5′-radiolabeling procedure (0.5 U/µL T4 PNK, 1X T4 PNK buffer, 1 µM RNA oligonucleotide, 500 nM [γ-$^{32}$P]-ATP, 2 hr at 37℃, 20 min at 65℃, buffer exchanged into water). A T4 RNA ligase 2 substrate was then formed by hybridizing the phosphate shuttle (363 nM) and the DNA oligo to be radiolabeled (303 nM) to a 'splint' RNA (333 nM) in annealing buffer (10 mM Tris-Cl, pH 7.9 at 25℃, 50 mM KCl, 1 mM EDTA) by heating to 95℃ and cooling to 25℃ over the course of 40 min on a thermocycler. T4 RNA ligase 2 (New England Biolabs) at 0.5 U/µL (manufacturer's units), 1X T4 RNA ligase 2 reaction buffer, and 1 mM MgCl$_2$ were added to this annealed structure. Ligation was allowed to proceed overnight at 37℃. The phosphate shuttle and splint RNA oligonucleotides were degraded by adding 150 mM NaOH and incubating at 95℃ for 10 min. The degradation reaction was stopped by adding a stoichiometric amount of HCl and placing on ice. The 3′-radiolabeled DNA oligonucleotide was then buffer-exchanged into 20 mM Tris-Cl (pH 7.9 at 25℃) using a Microspin G-25 spin column. This protocol has ~75% yield in terms of transfer of radioactivity from the phosphate shuttle RNA to the DNA oligonucleotide 3′ end. The hot hydroxide treatment causes slight accumulation of depurination products, but such products comprise a trivial fraction of the total population of radiolabeled DNA and do not interfere with downstream analysis. See *Supplementary file 1* for the identities and sequences of oligonucleotide reagents used in 3′-radiolabeling procedures.

## Permanganate reactivity experiments

The permanganate footprinting protocol was based on *Pul et al. (2012)*. In 40 µL permanganate reaction buffer (20 mM Tris-Cl, pH 7.9 at 25℃, 150 mM KCl, 5 mM MgCl$_2$), DNA (10 nM radiolabeled strand, 20 nM unlabeled strand), guide RNA or spacer mimic (100 nM), and protein (120 nM) were combined (omitting components as indicated for each experiment) and allowed to equilibrate to 30℃ for >5 min. 4 µL 160 mM KMnO$_4$ (solution prepared in permanganate reaction buffer immediately before reaction) was added and allowed to react for 2 min (unless otherwise indicated) at 30℃. Reactions were quenched with 4.8 µL β-mercaptoethanol and moved to ice. 5.3 µL 500 mM EDTA was added. 45.9 µL water was added, and samples were extracted once with 100 µL 25:24:1 phenol:chloroform:isoamyl alcohol (pH 8) in 5PRIME Phase Lock Heavy tubes (Quantabio). The aqueous phase was isolated and combined with 10 µL 3 M sodium acetate (pH 5.2), 1 µL GlycoBlue coprecipitant (Invitrogen), and 300 µL ethanol, and left at −20℃ for >2 hr. DNA was precipitated by centrifugation, and supernatant was decanted. Wet ethanol pellets were resuspended in 10% piperidine and incubated at 90℃ for 30 min. Solvent was evaporated in a SpeedVac (ThermoFisher Scientific). Approximate yield was determined by measuring radioactivity of the pellet-containing tube in a benchtop radiation counter (Bioscan QC-4000), and pellets were resuspended in an appropriate

volume of loading solution (50% water, 50% formamide, 0.025% w/v bromophenol blue) to normalize signal across samples prior to resolution by denaturing PAGE. Oligonucleotide identities and sequences are shown in *Supplementary file 1*.

## Denaturing polyacrylamide gel electrophoresis and phosphorimaging

Radiolabeled DNA oligonucleotides were denatured (95°C in 50% formamide for 3 min) and resolved on a denaturing polyacrylamide gel (15% acrylamide:bis-acrylamide 29:1, 7 M urea, 0.5X TBE). Gels were dried (4 hr, 80°C) on a gel dryer (Bio-Rad) and exposed to a phosphor screen. Phosphor screens were imaged on an Amersham Typhoon phosphorimager (GE Healthcare). Phosphorimages were quantified using ImageQuant software (GE Healthcare).

## Electrophoretic mobility shift assay and filter-binding assay

In both kinds of binding assays, complexes were formed in 1X binding buffer (20 mM Tris-Cl, pH 7.9 at 25°C, 150 mM KCl, 5 mM $MgCl_2$, 1 mM TCEP, 50 µg/mL heparin, 50 µg/mL bovine serum albumin, 5% glycerol). Cas protein was first diluted in series in binding buffer, added to a fixed concentration of guide RNA, and incubated at 37°C for 5 min, then 25°C for 25 min. This complex was then added to the radiolabeled DNA probe and incubated at 37°C for 5 min, then 25°C for 1 hr. When the titrant was crRNA instead of Cas protein, the Cas12a:crRNA complex was incubated at 25°C for 30 min, added to DNA probe, and incubated at 25°C for an additional 1 hr. For the EMSA, samples were then resolved on a native PAGE gel (8% acrylamide:bis-acrylamide 29:1, 0.5X TBE, 5 mM $MgCl_2$), which was dried and phosphorimaged. For the filter-binding assay, HT Tuffryn (Pall), Amersham Protran, and Amersham Hybond-N+ (GE Healthcare) membranes were equilibrated in 1X membrane wash buffer (20 mM Tris-Cl, pH 7.9 at 25°C, 150 mM KCl, 5 mM $MgCl_2$, 1 mM TCEP, 5% glycerol) and assembled on a vacuum dot-blot apparatus. Radioactive samples were applied to the membranes, and each spot was washed once with 40 µL 1X wash buffer. Membranes were air-dried and phosphorimaged. For assays testing complex assembly in calcium-containing buffer, 5 mM $CaCl_2$ was substituted for $MgCl_2$ in the binding buffer. Oligonucleotide identities and sequences are shown in *Supplementary file 1*.

## Enzymatic DNA cleavage assays

To initiate DNA cleavage, 1 vol of 2X surveillance complex, *trans*-active interference complex, or other nuclease (in 1X cleavage buffer) was combined with 1 vol of 2X radiolabeled DNA substrate (in 1X cleavage buffer) at 37°C (unless specified otherwise). For Cas12a, standard cleavage buffer was 10 mM Tris-Cl, pH 7.9 at 25°C, 150 mM KCl, 5 mM $MgCl_2$, 1 mM TCEP. 'Calcium-containing' cleavage buffer contained 5 mM $CaCl_2$ instead of $MgCl_2$. For Cas12i1, cleavage buffer was 50 mM Tris-Cl, pH 8.0 at 25°C, 50 mM NaCl, 10 mM $MgCl_2$. For S1 nuclease (ThermoScientific), cleavage buffer was the 1X reaction buffer provided by the manufacturer. At each timepoint, 1 vol of reaction was quenched with 1 vol of 2X quench buffer (94% formamide, 30 mM EDTA, 0.025% w/v bromophenol blue). For reactions catalyzed by Cas12i1, the 2X quench buffer also included 400 µg/mL heparin and 0.2% sodium dodecyl sulfate to prevent aggregation in gel wells. For 't = 0' timepoints, surveillance complex was first added to quench buffer and mixed, followed by addition of DNA substrate. Products were then resolved by denaturing PAGE and phosphorimaging. Oligonucleotide identities and sequences are shown in *Supplementary file 1*.

## 2-aminopurine fluorescence intensity measurements

All oligonucleotides used in these experiments were first ethanol-precipitated and resuspended to remove impurities from commercial synthesis that interfered with the optical spectra of interest. Oligonucleotides were combined to their final concentration in 1X nucleic acid spectroscopy buffer (10 mM $K_2HPO_4$/$KH_2PO_4$, pH 6.7, 150 mM KCl, 0.1 mM EDTA) and annealed on a thermocycler (95°C to 25°C over the course of 40 min). Final concentrations were 5 µM 2-AP-containing oligonucleotide, 5.5 µM complementary oligonucleotide (if present), 6 µM same-stranded oligonucleotide (if present), and 6 µM RNA oligonucleotide (if present). Samples were placed in a 1.5-mm fluorescence cuvette (Hellma Analytics) and allowed to equilibrate inside the temperature-controlled (30°C) cell of a QuantaMaster spectrofluorometer (Photon Technology International) for 3 min. The lamp power was set to 74 W, and the slit widths were set as follows (excitation slit 1: 0.5 mm; excitation slit 2: 1 mm;

emission slit 1: 2 mm; emission slit 2: 0.9 mm). Fluorescence intensity ($\lambda_{ex}$=310 nm (4 nm bandpass), $\lambda_{em}$=370 nm (3.6 nm bandpass)) was measured for 30 s, and the average across those 30 s was reported. Oligonucleotide identities and sequences are shown in *Supplementary file 1*.

## Dumbbell/hairpin melting temperature determination

All oligonucleotides used in these experiments were first ethanol-precipitated to remove impurities from commercial synthesis that interfered with the optical spectra of interest. Oligonucleotides were resuspended to an estimated 2.25 µM (the extinction coefficient of a highly stacked nucleic acid structure is difficult to estimate, but the unimolecular physical processes being probed are concentration-independent, in theory) in 1X nucleic acid spectroscopy buffer (10 mM $K_2HPO_4$/$KH_2PO_4$, pH 6.7, 150 mM KCl, 0.1 mM EDTA). Samples were placed in a 1-cm CD-grade quartz cuvette (Starna Cells) with a stir bar and cap, which was placed in the sample cell of a temperature-controlled spectrophotometer (Cary UV-Vis 100). An equivalent cuvette containing only nucleic acid spectroscopy buffer was placed in the reference cell. The stir apparatus was turned on, the block was heated to 95°C, and the samples were allowed to equilibrate for 3 min. The system was cooled to 2°C at 1 °C/minute, collecting an $A_{260}$ measurement every 0.5°C (averaging time = 2 s, slit bandwidth = 1 nm). Refolding of the Cas12a-like dumbbell at a slower temperature ramp rate (0.3 °C/minute) yielded results similar to those pictured, indicating that the faster ramp rate (1 °C/minute) was still slow enough that the absorbance measurements approached their equilibrium values. Oligonucleotide identities and sequences are shown in *Supplementary file 1*.

## Molecular dynamics simulations

The starting conformation of each junction was based on a relaxed structure of a chimeric RNA:RNA/DNA:DNA duplex. A 10-bp A-form RNA:RNA duplex and a 10-bp B-form DNA:DNA duplex were each built in x3DNA (*Lu and Olson, 2003*). These duplexes were manually placed in a coaxially stacked conformation using PyMOL (*DeLano, 2010*), and the junction was sealed on both strands using Coot (*Emsley et al., 2010*) (with slight adjustment of neighboring dihedrals to accommodate the introduced bonds). Using VMD (*Humphrey et al., 1996*), this system was solvated with TIP3 waters (*Jorgensen et al., 1983*) to a cube that stretched 15 Å past each edge of the nucleic acid duplex in its widest dimension. 150 mM NaCl and 5 mM $MgCl_2$ were added to the solvent using VMD. The system was minimized for 2000 steps with the nucleic acid atoms held fixed, then minimized for an additional 2000 steps while allowing all atoms to move. The system was then equilibrated for 1 ns, and the final structure of the chimeric duplex was used as the basis for building the two kinds of junctions.

For the RNA-3′ junction, the bond between the internal 3′ end of the RNA tract and the internal 5′ end of the DNA tract was removed, and a 2-nt DNA flap was modeled in an arbitrary conformation. On the opposite strand, all ribonucleotides were changed to 2′-deoxyribonucleotides, and uracils were changed to thymines. The RNA-5′ junction was built analogously. For both junctions, the outer termini of each duplex contained 5′-OH and 3′-OH. The internal DNA flap contained a 5′-phosphate (RNA-3′ junction) or a 3′-OH (RNA-5′ junction), in keeping with the chemical products of RuvC-catalyzed DNA cleavage. The internal RNA end contained a 3′-OH (RNA-3′ junction) or a 5′-OH (RNA-5′ junction). These systems were solvated and minimized as before. The systems were then equilibrated for 1 ns with the nucleic acid atoms held fixed. This system served as the starting state for 10 separate production trajectories that were each run for 50 ns with all atoms free. All equilibration and production runs were carried out in the NPT ensemble at a temperature of 300 K and pressure of 1 atm.

The simulations were performed on XSEDE computing resources (*Towns et al., 2014*) using the NAMD (*Phillips et al., 2005*) package with the CHARMM36m forcefield (*Huang et al., 2017*) and an integration timestep of 2 fs. The Particle Mesh Ewald approximation was used to calculate long-range electrostatic interactions (*Darden et al., 1993*) with a grid size of 1 Å. Van der Waals interactions were truncated at 12 Å. Hydrogen atoms bonded to heavy atoms were constrained with the ShakeH algorithm (*Ryckaert et al., 1977*). The Langevin thermostat was used to control the temperature with a damping coefficient of 1/ps, applied to non-hydrogen atoms. Pressure was controlled with the Nose-Hoover Langevin method (*Martyna et al., 1994*; *Feller et al., 1995*), with a Langevin piston period of 200 fs and a piston decay time of 50 fs.

For each trajectory, the coordinates of the two nucleobases at the junction on the flapped strand were isolated for further analysis at a sampling rate of 1/ps. Envelope surface area (ESA), defined as the solvent-exposed surface area of the two isolated nucleobases, was determined in PyMOL and serves as a metric of the degree of base stacking (bases that are well-stacked have a low ESA, whereas bases that are unstacked have a high ESA). All figures were prepared in PyMOL.

## DNA size standard preparation

To identify a known specific cleavage site within the protospacer, radiolabeled DNA duplexes were digested with TseI (New England Biolabs) (0.025 U/μL final concentration, manufacturer's units) in DNase buffer (10 mM Tris-Cl, pH 7.9 at 25°C, 150 mM KCl, 5 mM MgCl₂, 1 mM TCEP) for 10 min at 65°C. To generate a single-nucleotide ladder, the same radiolabeled DNA oligonucleotides were separately digested with nuclease P1, DNase I, *trans*-active AsCas12a, and T5 exonuclease (which all leave 5′ phosphate and 3′-OH on their cleavage products, chemically equivalent to products of Cas12a *cis* cleavage), and products were pooled at a 1:1:1:1 ratio (T5 exonuclease was not used for 5′-radiolabeled oligonucleotides) prior to loading on the gel. Nuclease P1 (New England Biolabs) digests were performed with single-stranded radiolabeled DNA oligonucleotide and 0.5 U/μL enzyme (manufacturer's units) in DNase buffer for 3 min at 37°C. DNase I (RQ1 RNase-Free DNase, Promega) digests were performed with radiolabeled DNA duplex and 0.01 U/μL enzyme (manufacturer's units) in DNase buffer for 3 min at 37°C. AsCas12a digests were performed with single-stranded radiolabeled DNA oligonucleotide, 100 nM AsCas12a, 120 nM crRNA, and 50 nM pre-cleaved DNA activator in DNase buffer for 5 min at 37°C. T5 exonuclease (New England Biolabs) digests were performed with radiolabeled DNA duplex and 0.01 U/μL enzyme (manufacturer's units) in DNase buffer for 10 min at 37°C. All reactions were stopped by addition of 1 vol 2X quench buffer (94% formamide, 30 mM EDTA, 0.025% w/v bromophenol blue).

## Steady-state Cas12a *trans* DNA cleavage kinetic analysis

Kinetics of ssDNA cleavage were assessed by monitoring the rate of dequenching of a fluorophore-DNA-quencher substrate, as in *Chen et al. (2018)*. Briefly, *trans*-active holoenzyme (final concentrations: 100 nM AsCas12a, 120 nM crRNA, 10 nM pre-cleaved duplex activator) was added to various concentrations of fluorophore-DNA-quencher substrate, and fluorescence (excitation filter: 485 nm/20 bandpass, emission filter: 528 nm/20 bandpass) was monitored over time at 37°C on a Cytation5 fluorescence plate-reader (BioTek). $V_0$ was determined as (slope$_{all\ components}$ - slope$_{no\ DNA\ activator}$)× (fluorescence intensity:[product] conversion factor). The fluorescence intensity:[product] conversion factor was determined empirically for each separate concentration of fluorescent reporter (by equilibrium titration of purified cleaved/uncleaved reporter), as the relationship departed from linearity at higher substrate concentrations. Oligonucleotide identities and sequences are shown in *Supplementary file 1*.

## Model fitting

All models were fit by the least-squares method in Prism 7 (GraphPad Software). The model used for each dataset is described in the corresponding figure legend.

## Analysis and interpretation of permanganate reactivity data

In this work, data describing permanganate reactivity are presented in three ways:

1. Raw phosphorimages of denaturing PAGE analysis of DNA substrates treated with permanganate and piperidine.
2. 'Permanganate reactivity index' (PRI) of individual thymine nucleobases. This metric is determined from the raw phosphorimages. It is an approximation of the absolute rate of oxidation at a given thymine, linearly normalized such that PRI = 1 describes a thymine that is fully single-stranded. Thus, a thymine with PRI = 0.4 is estimated to have been oxidized twice as fast as a thymine with PRI = 0.2.
3. 'Fraction oxidized in A/T tract' (FO). This metric is a mathematical transformation/combination of the PRI of all thymines in the R-loop flank of a given DNA substrate. It is an approximation of the total fraction of DNA molecules (the two strands of DNA forming the R-loop flank are referred to here as a single 'molecule') that, at the moment of quenching, have been oxidized on at least one of the nine thymines within the R-loop flank.

While visually inspecting phosphorimages from permanganate experiments, note that there are occasionally faint bands corresponding to strand cleavage at cytosines (permanganate oxidizes cytosines, albeit much more slowly than thymines) and at purines (which occurs during hydroxide treatment in the 3′-radiolabeling protocol). Such bands constituted a trivial fraction of the total lane volume and did not meaningfully affect analysis.

Additionally, visual inspection of the raw phosphorimages can be informative but should be approached with caution because the absolute volume of a given band is meaningless without considering other bands that may have detracted from its signal. For example, a strongly oxidized thymine (thymine 1) may yield only a faint band if another strongly oxidized thymine (thymine 2) lies between thymine 1 and the radiolabeled terminus of the DNA oligonucleotide. If all thymine oxidation events are independent of each other (i.e., thymine 1 has the same oxidation probability irrespective of whether thymine 2, or any other thymine, has been oxidized or not), the oxidation probability of thymine 1 can be reconstructed by considering the thymine-1 band only as a subpopulation of the bands above it on the gel. In other words, out of all the DNA molecules on the gel for which oxidation of thymine 1 would have been observable (i.e., cleavage fragments *at or above* the thymine-1 fragment on the gel), what fraction of those molecules *were* in fact oxidized at thymine 1? In reality, clusters of thymines have been observed to mutually enhance oxidation probability (*Nomura and Okamoto, 2008*), so perfect independence *cannot* be assumed. Thus, the parameters described below are imperfect measures of the true rate of oxidation at each thymine.

Beyond uncertainty in the measurement, it is also unknown to what extent the probing technique is changing the fundamental biophysical features of the DNA structures. Notably, thymine's reaction with permanganate breaks the planarity of the nucleobase and, consequently, its capacity to stack normally. In an A/T-rich sequence like our R-loop flank (*Figure 2B*), an oxidation event at thymine 1 could, in principle, begin a chain reaction of oxidation events as each adjacent thymine successively loses planarity and unstacks, exposing its neighbor to the oxidant. If such chain reactions occurred quickly as compared to the timescale of the assay (2 min), the distribution of band volumes would be skewed toward thymine 9. In reality, the band volume distributions are skewed sharply toward thymine 1 (*Figure 2—figure supplement 3*), suggesting that, on the assayed timescale, the majority of oxidation events do *not* lead to additional oxidation events. Still, the possibility of chain reactions should be kept in mind when interpreting the observed permanganate reactivity patterns, in which reactivity decreases with distance from the R-loop edge (*Figure 2B*). While these patterns are consistent with fraying duplex termini, the apparent 'depth' of the fraying events should be interpreted as an upper limit on what would occur in a substrate unexposed to permanganate.

Finally, the structural determinants of permanganate reactivity should be considered carefully when using these data to draw conclusions about DNA conformation. While high permanganate reactivity is often associated with 'single-strandedness' or 'lack of base pairing,' the reaction is more precisely dependent upon the ability of a permanganate molecule to approach the C5=C6 bond of the thymine nucleobase. This approach could be facilitated by assumption of a non-B-form helical geometry, global melting of the DNA duplex, or 'flipping' of a thymine out of the duplex without dramatically affecting the helical geometry (*Bui et al., 2003*). Furthermore, a thymine lying on a duplex terminus could, in principle, be approached and attacked while base paired, albeit from a restricted angle. This possibility is especially important to consider for thymine 1 of our A/T-rich R-loop flank (*Figure 2B*). The reactivity of this thymine varies in RNA-free DNA bubble controls that have different bubble sequences (*Figure 4—figure supplement 3*), perhaps reflecting differences in the propensities of individual (unpaired) neighboring bases to stack on the duplex-terminal thymine. Finally, because thymines within the RNA:DNA hybrid of R-loop structures have two possible base pairing partners (DNA versus crRNA), the conformational ensemble at these positions is highly complex, and we did not attempt to draw any structural conclusions from their oxidation rates.

Given the aforementioned caveats, the PRI and FO metrics described below should be interpreted as estimates rather than accurate measurements of rate and extent of reaction. Additionally, permanganate reactivity data should be considered alongside the orthogonal techniques used in this work to assess the structure, energetics, and conformational dynamics of interhelical junctions. The definitions of permanganate reactivity index (PRI) and fraction oxidized (FO) are as follows:

Let $v_i$ denote the volume of band $i$ in a lane with $n$ total bands (band 1 is the shortest cleavage fragment, band $n$ is the topmost band corresponding to the starting/uncleaved DNA oligonucleotide). The probability of oxidation at thymine $i$ is defined as:

$$p_i = \frac{v_i}{\sum\limits_{j=i}^{n} v_j}$$

Note that this relationship allows determination of $p_i$ even if the values of $p_{1 \leq x < i}$ are unavailable (e.g., if the shortest cleavage products have been run off the bottom of the gel). Assuming thymine oxidation occurs with a uniform probability across the time course of permanganate application (see exponential curve in *Figure 2—figure supplement 3*) the rate constant associated with oxidation probability $p_i$ is defined as:

$$k_i = \frac{\ln\left(\frac{1}{1-p_i}\right)}{t}$$

where $t$ is the time of quenching. We found that across experimental replicates there was systematic variation in $k$ (e.g., $k$ was universally smaller in replicate 2 than in replicate 1 for any given thymine), likely due to variability in the oxidation activity of each new preparation of the potassium permanganate solution. To allow comparison across replicates, we normalized all values of $k$ to that of a reference thymine ($k_{ref}$) whose conformational dynamics were not expected to be affected by R-loop formation or associated substrate variations (the thymine 10 nt from the end of the 5′-radiolabeled oligo, present in all DNA substrates tested). For every set of replicate experiments, which each involved a new preparation of potassium permanganate solution, we determined the average rate constant of the reference thymine across all substrates ($\bar{k}_{ref}$). The global average of $k_{ref}$ across all experiments and all replicates ($\mu_{ref}$) was taken to be the true value of $k_{ref}$. The corrected value of $k_i$ for each thymine was then taken to be:

$$k_{i,corr} = \frac{\mu_{ref}}{\bar{k}_{ref}} k_i$$

The permanganate reactivity index was then calculated as:

$$PRI_i = \frac{k_{i,corr}}{k_{ss,corr}}$$

where $k_{ss,corr}$ is the reference-corrected oxidation rate constant for a thymine unassociated with a stable base-pairing partner. The value of $k_{ss,corr}$ used in our calculations was 0.79 min$^{-1}$, empirically determined for an arbitrarily chosen thymine within a DNA bubble (*Figure 2—figure supplement 3*). The estimated fraction of DNA molecules oxidized on at least one thymine within the R-loop flank (correcting to the value expected if the potassium permanganate solution had its average oxidation activity) was then calculated as:

$$p_{i,corr} = 1 - e^{-k_{i,corr}t}$$

$$FO = 1 - \prod_{a \in RLF} (1 - p_{a,corr}) = 1 - \exp\left[-\left(\sum_{a \in RLF} k_{a,corr}\right)t\right]$$

where $RLF$ denotes the set of band indices corresponding to the thymines labeled $T_1$ through $T_9$ in *Figure 2B*, combining data from both the NTS-radiolabeled and TS-radiolabeled experiments. Note that the PRI metric is subject to increased uncertainty as $p_i$ approaches 1, where the slope of $\ln\left(\frac{1}{1-p_i}\right)$ approaches infinity.

## Acknowledgements

We thank the lab of Andreas Martin for use of their temperature-controlled spectrofluorometer. We thank Gavin Knott, Andreas Martin, David Wemmer, and Kevan Shokat for helpful discussions. We

thank Lucas Harrington and Janice Chen for a critical reading of the manuscript. This work used the Extreme Science and Engineering Discovery Environment (XSEDE) (Stampede2, Texas Advanced Computing Center, allocation MCB170063), which is supported by National Science Foundation grant number ACI-1548562.

## Additional information

### Competing interests

John Kuriyan: Senior Editor, eLife. Isaac P Witte: I.P.W. served as a consultant for Mammoth Biosciences. Jennifer A Doudna: The Regents of the University of California have patents issued and pending for CRISPR technologies on which J.A.D. is an inventor. J.A.D. is a cofounder of Caribou Biosciences, Editas Medicine, Scribe Therapeutics, and Mammoth Biosciences. J.A.D. is a scientific advisory board member of Caribou Biosciences, Intellia Therapeutics, eFFECTOR Therapeutics, Scribe Therapeutics, Mammoth Biosciences, Synthego, Felix Biosciences, and Inari. J.A.D. is a Director at Johnson & Johnson and has research projects sponsored by Biogen, Pfizer, AppleTree Partners, and Roche. The other authors declare that no competing interests exist.

### Funding

| Funder | Grant reference number | Author |
| --- | --- | --- |
| Howard Hughes Medical Institute | Doudna Lab | Jennifer A Doudna |
| National Science Foundation | MCB-1817593 | Jennifer A Doudna |
| Howard Hughes Medical Institute | Kuriyan Lab | John Kuriyan |
| National Science Foundation | Graduate Research Fellowship | Joshua C Cofsky |

The funders had no role in study design, data collection and interpretation, or the decision to submit the work for publication.

### Author contributions

Joshua C Cofsky, Conceptualization, Data curation, Formal analysis, Validation, Investigation, Methodology, Writing - original draft, Writing - review and editing; Deepti Karandur, Carolyn J Huang, Isaac P Witte, Investigation, Methodology; John Kuriyan, Conceptualization, Supervision, Project administration, Writing - review and editing; Jennifer A Doudna, Conceptualization, Supervision, Writing - original draft, Project administration, Writing - review and editing

### Author ORCIDs

Joshua C Cofsky https://orcid.org/0000-0001-5403-8555
Isaac P Witte https://orcid.org/0000-0002-3879-0306
John Kuriyan http://orcid.org/0000-0002-4414-5477
Jennifer A Doudna https://orcid.org/0000-0001-9161-999X

### Decision letter and Author response

Decision letter https://doi.org/10.7554/eLife.55143.sa1
Author response https://doi.org/10.7554/eLife.55143.sa2

## Additional files

### Supplementary files

• Supplementary file 1. Sequences of plasmids, DNA oligonucleotides, and RNA oligonucleotides used in this work.

• Transparent reporting form

## Data availability

All data generated or analyzed during this study are included in the manuscript and supporting files, with the exception of raw phosphorimages (which are difficult to interpret without the authors' guidance because of complex and arbitrary gel-loading schemes) and fully sampled molecular dynamics trajectories (which have a file size of ~2.2 TB). Representative images and trajectories have been included in the manuscript, supplementary figures, and/or source data. The original files are available from the authors upon request.

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

## Appendix 1

### The RuvC active site of Cas12a selectively cuts single-stranded DNA

While the ssDNA specificity of Cas12a RuvC *trans* activity has been previously reported (*Chen et al., 2018*), we wanted to probe the strength and mechanism of this specificity, which could lend insight into the enzyme-substrate conformations that permit *cis* cleavage. For example, does *trans*-active Cas12a only cut DNA substrates that are completely single-stranded, or will it also cut small distortions in dsDNA, as has been observed for S1 nuclease, an unrelated DNase with reported specificity for single-stranded substrates (*Wiegand et al., 1975*)? To compare the substrate range of Cas12a RuvC to that of S1 nuclease, we tested the susceptibility of various radiolabeled DNA structures (including a single strand, a duplex, a nicked duplex, and duplexes with gaps, bubbles, and bulges) to cleavage by the two enzymes, used at concentrations with comparable activity. In contrast to S1 nuclease, which exhibited minimal discrimination against even the fully duplex substrate, Cas12a discriminated strongly against substrates with up to 8-nt tracts of non-duplex DNA. This stringent substrate preference suggests that non-ssDNA structures are either sterically excluded from the RuvC active site or unable to assume catalytic geometry once bound. However, strand discontinuities as small as a nick were sufficient to permit low levels of internal strand cleavage by Cas12a (*Appendix 1—figure 1*, *Appendix 1—figure 1—figure supplement 1*). Notably, the nicked structure resembles the juxtaposed duplexes of an R-loop boundary, and the increased sensitivity of this substrate to *trans* cleavage may emerge from the same phenomenon that enables *cis* cleavage of the target strand.

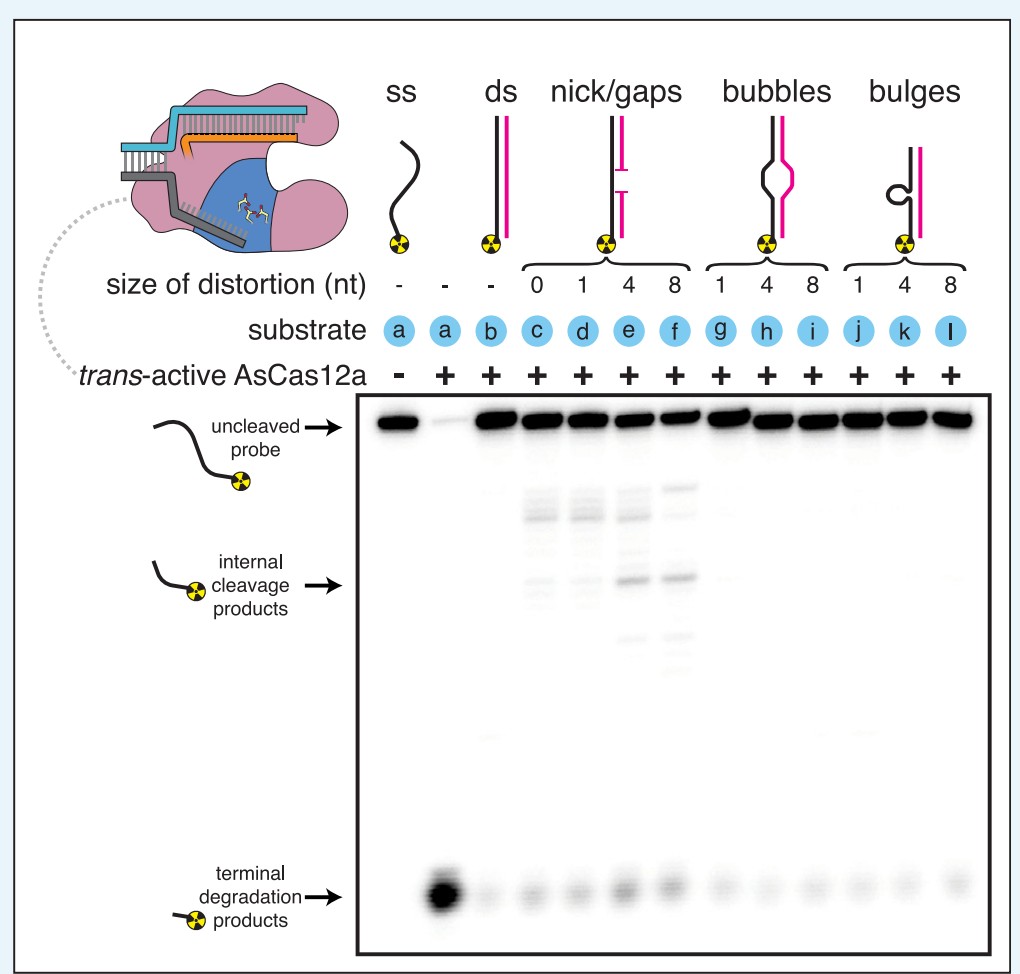

**Appendix 1—figure 1.** Substrate specificity of Cas12a *trans*-active holoenzyme. Phosphorimage of AsCas12a cleavage products, resolved by denaturing PAGE. *Trans*-active AsCas12a holoenzyme (115 nM of each component: protein, crRNA, pre-cleaved activator) was incubated with 1 nM of the indicated substrate for 2 hours at 30°C prior to quenching. Substrate (a) was a single-stranded DNA oligonucleotide with no homology to the crRNA. To generate substrates (b) through (l), substrate (a) was hybridized to a variety of unlabeled complementary DNA oligonucleotides. Substrate (c) contained a nick. Substrates (d), (e), and (f) contained gaps of 1, 4, and 8 nt, respectively. Substrates (g), (h), and (i) contained bubbles of 1, 4, and 8 nt, respectively. Substrates (j), (k), and (l) contained bulges of 1, 4, and 8 nt, respectively.

The online version of this article includes the following source data and figure supplement(s) for figure 1:

**Appendix 1—figure 1—source data 1.** Numerical data plotted in *Appendix 1—figure 1—figure supplement 1*. **Appendix 1—Figure 1 supplement 1.** Comparing the substrate specificities of Cas12a *trans*-active holoenzyme and S1 nuclease.

## Appendix 2

# Non-target-strand gap formation is required for efficient cleavage of the target strand

Elucidating the mechanism of target-strand cleavage requires an understanding of its interplay with non-target-strand cleavage. To precisely determine the location of Cas12a-catalyzed NTS cleavage, we monitored the formation of DNA cleavage products over time by denaturing PAGE, and we distinguished between different DNA fragments by placing radiolabels on the 5′ or 3′ end of each strand (*Appendix 2—figure 1*, *Appendix 2—figure 1—figure supplement 1*, *Figure 2—figure supplement 1*). These experiments were conducted on a timescale and at an enzyme concentration for which *cis* cleavage events (i.e., events in which a Cas12a molecule cuts the DNA molecule to whose PAM it is bound) are the primary contributor to the observed DNA cutting signal (*Appendix 2—figure 1—figure supplements 2, 3*). Additionally, while *trans* cleavage events (i.e., events in which a Cas12a molecule cuts free DNA or DNA bound to another Cas12a molecule) may minorly contribute, the concentration-dependence of the *trans* cleavage mode allows it to be distinguished from *cis* cleavage processes (*Appendix 2—figure 1—figure supplements 4, 5*).

According to these mapping experiments, the non-target strand has two major cleavage sites, at dinucleotides 13/14 and 18/19 (numbers denote distance from the PAM), suggesting the formation of a 5-nt gap within the tract of DNA displaced by the crRNA (*Appendix 2—figure 1B*). The evolution of the cleavage pattern over time indicates that, in at least some fraction of the molecules assayed, the NTS is first cut between the two major sites and achieves its final state through two or more 'trimming' events. A Cas12a ortholog from *Francisella novicida* also produced a gap in the NTS, implying that this phenomenon may be conserved across type V-A enzymes (*Appendix 2—figure 1—figure supplement 6*).

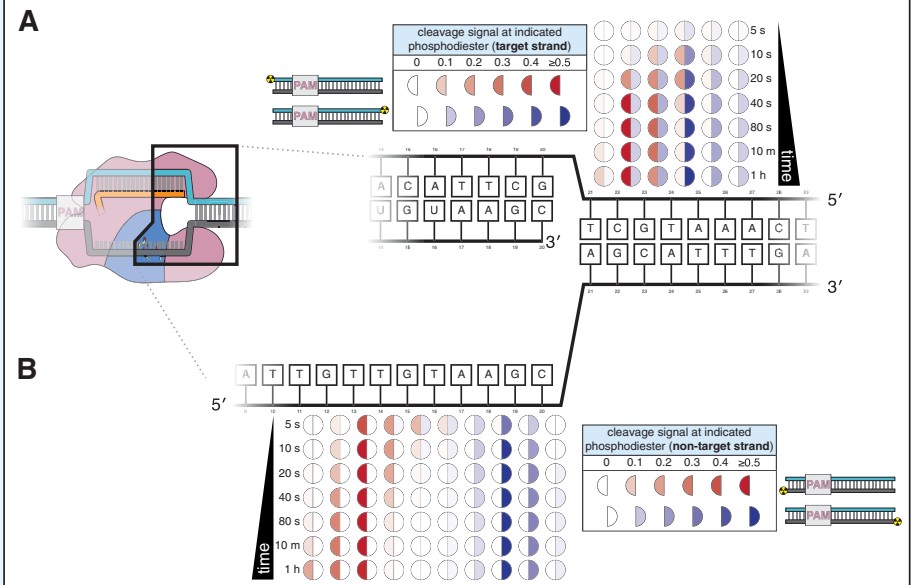

**Appendix 2—figure 1.** Cas12a forms a gap in the non-target strand and cleaves the target strand outside the R-loop. (**A**) Target-strand cleavage products over time, as quantified by denaturing PAGE. 100 nM AsCas12a and 120 nM crRNA were incubated with 5 nM radiolabeled DNA target at 37°C for the indicated timepoints, followed by quenching and resolution by denaturing PAGE. Representative phosphorimages are shown in *Appendix 2—figure 1—figure supplement 1*. Data shown here are the average of three replicates. Each circle denotes a phosphodiester at which cleavage was observed. The intensity of color in each

half-circle ('cleavage signal') reflects the fraction (band volume for a given cleavage product) / (total volume in lane). The left half of each circle (red) corresponds to the cleavage product detected with a PAM-proximal radiolabel. The right half of each circle (blue) corresponds to the cleavage product detected with a PAM-distal radiolabel. (**B**) Non-target-strand cleavage products over time, as quantified by denaturing PAGE (phosphorimage in *Appendix 2—figure 1—figure supplement 1*). Data representation as in **A**.

The online version of this article includes the following source data and figure supplement(s) for figure 1:

**Appendix 2—figure 1—source data 1.** Numerical data plotted in *Appendix 2—figure 1* and accompanying figure supplements. **Appendix 2—Figure 1 supplement 1.** Denaturing PAGE analysis of AsCas12a cleavage products.
**Appendix 2—Figure 1 supplement 2.** Steady-state kinetic analysis of AsCas12a *trans* DNase activity, as measured by fluorophore dequenching.
**Appendix 2—Figure 1 supplement 3.** Enzyme-concentration dependence of AsCas12a *cis* DNA cleavage kinetics.
**Appendix 2—Figure 1 supplement 4.** Concentration dependence of various modes of DNase activity.
**Appendix 2—Figure 1 supplement 5.** Concentration dependence of phosphodiester-mapped cleavage events.
**Appendix 2—Figure 1 supplement 6.** Cleavage product mapping for FnCas12a.

Because our ensemble biochemical assay is blind to the occurrence of additional cuts that occur farther from the radiolabel than the first cut, we cannot unambiguously assign cleavage states to individual interference complex molecules. Nevertheless, by comparing the 5'- and 3'-mapped cut-site distributions at a given timepoint, we can roughly assess the predominant cleavage state of individual molecules. For example, a population of DNA molecules cut exactly once would yield completely overlapping 5'-/3'-mapped cut-site distributions, while a population with a gap would yield non-overlapping peaks in the two distributions.

Therefore, in the target-strand mapping experiments, overlap of the 5'- and 3'-mapped distributions is consistent with (although not uniquely explainable by) a population of interference complexes that have cleaved the TS exactly once (*Appendix 2—figure 1A*). However, separation of the two distribution peaks (at dinucleotides 22/23 and 24/25) indicates that most individual complexes perform at least one additional cut in the TS, yielding a small TS gap prior to dissociation of the PAM-distal cleavage product (*Appendix 2—figure 1A*; *Singh et al., 2018*).

These results show that while Cas12a activity does generate staggered cuts in dsDNA targets as previously reported (*Zetsche et al., 2015*), its trimming activity, most notably on the NTS, destroys the capacity of released DNA products to be trivially re-ligated in a restriction-enzyme-like procedure. In this particular DNA target, the major Cas12a cleavage products contain a 9-nt overhang on the PAM-proximal fragment and a 6-nt overhang on the PAM-distal fragment, with 4 nt of complementarity between these overhangs (*Appendix 2—figure 1*). These findings may aid in the study of eukaryotic DNA repair pathways elicited by Cas12a cleavage events and in biotechnological applications of Cas12a that exploit its staggered cuts.

For Cas12a, non-target-strand cleavage is a prerequisite of target-strand cleavage (*Swarts and Jinek, 2019*; *Appendix 2—figure 2—figure supplement 1*). To test whether gap formation in the NTS is also an obligatory step of Cas12a-catalyzed DNA cleavage, we modified the NTS with phosphorothioates to selectively inhibit cleavage of certain DNA linkages. Substitution of calcium for magnesium in the reaction buffer produced a 13-fold increase in the RuvC DNase's selectivity for phosphodiester over phosphorothioate linkages, allowing us to effectively halt expansion of a defined gap size in the NTS while simultaneously measuring cleavage of a fully phosphodiester-linked TS (*Appendix 2—figure 2—figure supplement 2*). Neither the calcium substitution nor the inclusion of phosphorothioates impaired enzyme-substrate complex assembly (*Appendix 2—figure 2—figure supplement 3*).

Using a series of NTS variants that were chemically locked in various states of cleavage (i.e., intact, a single nick, and gaps of varying sizes) (*Appendix 2—figure 2—figure supplement 4*), we measured the extent of TS cleavage after one hour in calcium-containing buffer. TS cleavage was almost undetectable in the presence of a single nick, and its extent of cleavage

only reached that observed with a fully phosphodiester-linked NTS when the gap was widened to 5 nt (*Appendix 2—figure 2A*, *Appendix 2—figure 2—figure supplement 5*). We observed a similar trend when the experiment was conducted in the presence of magnesium (*Appendix 2—figure 2—figure supplement 6*). Together, these results indicate that formation of a gap in the NTS accelerates TS cleavage. Although NTS gap formation is not strictly required for TS cleavage to occur, our bulk cleavage analysis suggests that the NTS gap does in fact form before TS cleavage in the native Cas12a cleavage pathway (*Appendix 2—figure 1*). Thus, for most experiments in this work that probed the mechanism of TS cleavage (*Figures 2–5*), we used substrates that recapitulated the 5-nt NTS gap (referred to as a 'pre-cleaved' or 'pre-gapped' NTS).

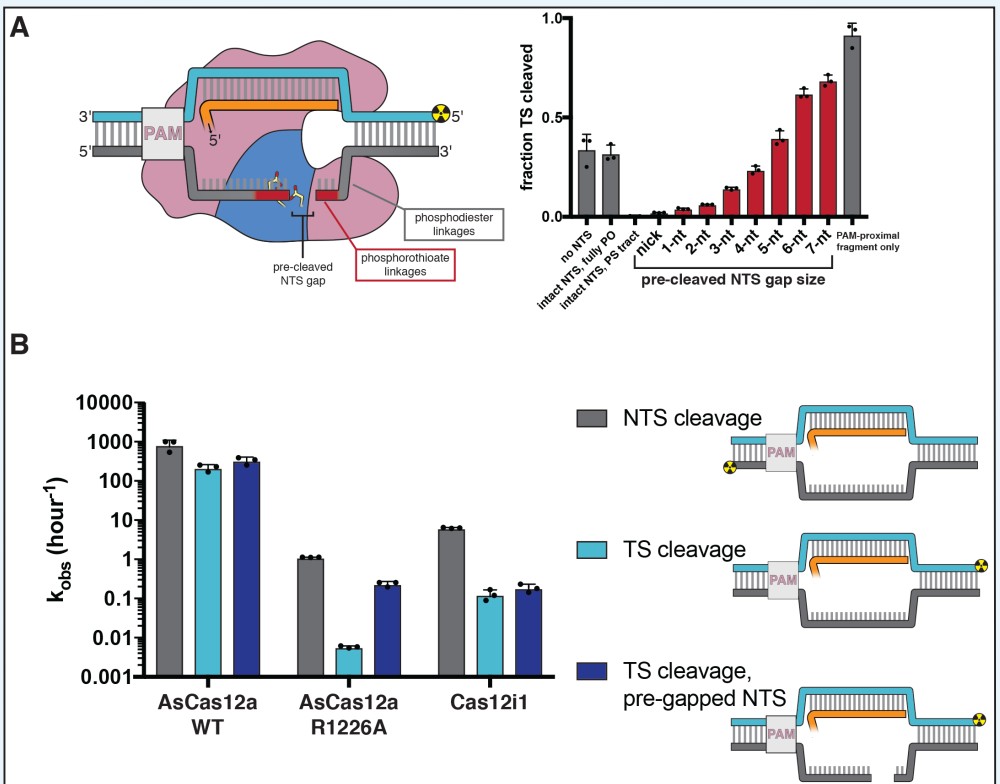

**Appendix 2—figure 2.** Non-target-strand gap formation poses a kinetic barrier to target-strand cleavage for AsCas12a. (**A**) Extent of target-strand cleavage by wild type AsCas12a in the presence of various non-target-strand variants, as resolved by denaturing PAGE (phosphorimage in *Appendix 2—figure 2—figure supplement 5*). Cas12a surveillance complex (100 nM AsCas12a, 120 nM crRNA) was added to 1 nM pre-hybridized target DNA radiolabeled on the 5' end of the TS and allowed to incubate in cleavage buffer with 5 mM $CaCl_2$ for 1 hr at 37°C prior to quenching. In the schematic, the red portion of the NTS denotes phosphorothioate (PS) linkages; the gray portion denotes phosphodiester (PO) linkages. In the graph, red bars denote reactions with a PS-containing NTS variant; gray bars denote reactions either with no NTS or with an NTS variant containing only PO-linkages. From left to right (omitting the no-NTS control), the NTS variants used were A, B, D, G, J, N, Q, T, W, Y, Z, as schematized in *Appendix 2—figure 2—figure supplement 4*. Columns and associated error bars indicate the mean and standard deviation of three replicates. (**B**) Cleavage kinetics of NTS, TS, and TS complexed with a pre-gapped NTS (NTS contains a 5-nt gap). 100 nM protein and 120 nM cognate crRNA were incubated with 2 nM DNA target with a 5' radiolabel on the indicated strand at 37°C for various timepoints, followed by quenching and resolution by denaturing PAGE. Representative phosphorimages and quantifications are

shown in *Appendix 2—figure 2—figure supplement 8*. Columns and associated error bars indicate the mean and standard deviation of three replicates.

The online version of this article includes the following source data and figure supplement(s) for figure 2:

**Appendix 2—figure 2—source data 1.** Numerical data plotted in *Appendix 2—figure 2* and accompanying figure supplements. **Appendix 2—Figure 2 supplement 1.** Non-target-strand cleavage precedes target-strand cleavage for AsCas12a and Cas12i1.
**Appendix 2—Figure 2 supplement 2.** Cleavage at phosphorothioates can be selectively slowed by substitution of $CaCl_2$ for $MgCl_2$.
**Appendix 2—Figure 2 supplement 3.** Interference complexes are stable in the presence of $CaCl_2$ and with a phosphorothioated DNA target.
**Appendix 2—Figure 2 supplement 4.** Non-target-strand variants used in gap-dependence experiments.
**Appendix 2—Figure 2 supplement 5.** Phosphorimage and quantification of non-target-strand gap-dependence experiments, in $CaCl_2$.
**Appendix 2—Figure 2 supplement 6.** Phosphorimage and quantification of non-target-strand gap-dependence experiments, in $MgCl_2$.
**Appendix 2—Figure 2 supplement 7.** Phosphorimage and quantification of non-target-strand gap-dependence experiments, in $MgCl_2$, with radiolabeled *trans* substrate.
**Appendix 2—Figure 2 supplement 8.** Phosphorimages and quantification of pre-gapped non-target-strand experiments.
**Appendix 2—Figure 2 supplement 9.** Affinity measurements for RNA-guided interaction of AsCas12a mutants with dsDNA.
**Appendix 2—Figure 2 supplement 10.** Non-target-strand cleavage product mapping for AsCas12a R1226A.
**Appendix 2—Figure 2 supplement 11.** Cleavage product mapping for Cas12i1.

The dependence of TS cleavage on NTS gap formation suggests that the NTS occludes the RuvC active site immediately following R-loop formation. After a NTS nicking event, RuvC releases and rebinds the NTS in different registers to cleave it in multiple locations, forming a gap that clears the active site for entry of the TS. Consistent with a substrate-occlusion model, we determined that *trans* ssDNA cleavage is enhanced by gap formation in the NTS and only achieves its maximal rate when the TS has also been cut (*Appendix 2—figure 2—figure supplement 7*). These observations hint at a possible evolutionary origin for non-specific *trans* activity—an enzyme that must loosely shuttle multiple *cis* substrates into and out of a shared catalytic center would benefit from a promiscuous and 'open' active site. Therefore, the target-activated non-specific ssDNase activity of type V Cas enzymes may be a mechanistic artifact of single-DNase *cis* cleavage rather than a direct immunological necessity.

To further understand the interplay between non-target-strand and target-strand cleavage, we investigated strand cleavage kinetics in two type V Cas enzymes that have been reported to act as NTS 'nickases' (i.e., cleave the NTS but not the TS): the R1226A mutant of AsCas12a (R1226 lies adjacent to the RuvC active site in the tertiary protein structure) (*Yamano et al., 2016*) and the type V-I interference enzyme Cas12i1 (*Yan et al., 2019*). With our sensitive phosphorimaging assay, we determined that these enzymes do in fact cleave the TS, albeit slowly, suggesting that the cleavage signal simply did not rise above the detection limit in previous experiments (*Yamano et al., 2016*; *Yan et al., 2019*; *Appendix 2—figure 2B*, *Appendix 2—figure 2—figure supplement 8*). Importantly, slow TS cleavage activity was coupled to slow NTS cleavage activity for both AsCas12a R1226A and Cas12i1 (*Appendix 2—figure 2B*, *Appendix 2—figure 2—figure supplement 8*), suggesting that slow TS cleavage emerges mostly from low overall catalytic efficiency. While this low efficiency could be explained by weak target association for Cas12i1, which exhibited no detectable DNA binding activity in our filter-binding assay, the affinity of AsCas12a R1226A for DNA was unimpaired as compared to WT AsCas12a (*Appendix 2—figure 2—figure supplement 9*). Still, both enzymes exhibited $k_{TS}:k_{NTS}$ ratios lower than that of WT AsCas12a, suggesting that there may also be more fundamental differences in their DNA cleavage pathways (*Appendix 2—figure 2B*).

Specifically, we wondered whether these enzymes were able to form non-target-strand gaps. To test this question, we began by performing cleavage site mapping on the NTS of

AsCas12a R1226A. At the 1-hr timepoint, the 5′- and 3′-mapped cut-site distributions contained significant overlap, suggestive of a population of DNA strands containing either a single nick or a small gap (*Appendix 2—figure 2—figure supplement 10*). In contrast, at the 5-s timepoint of the NTS mapping experiments for WT AsCas12a, the two distributions were almost completely non-overlapping, having already developed peaks at the major cut sites (*Appendix 2—figure 1B*, *Appendix 2—figure 2—figure supplement 10*). Because the 1-hr timepoint for AsCas12a R1226A and the 5-s timepoint for WT AsCas12a have similar values (~70%) of total NTS cleavage (an unambiguous measure of the fraction of molecules that have experienced at least one cut), the difference in distributions implies a fundamental difference across the two enzymes in terms of their relative rates of NTS nicking and trimming (i.e., R1226A has a lower ratio of $k_{trim}$:$k_{nick}$ than WT) (*Appendix 2—figure 1B*, *Appendix 2—figure 2—figure supplement 10*).

To more directly probe the kinetic contribution of NTS trimming activity, we measured the rate of AsCas12a R1226A TS cleavage in an interference complex with a pre-gapped NTS. The observed TS cleavage rate in this complex was 40-fold higher than in the one with an intact NTS (*Appendix 2—figure 2B*), suggesting that the physical basis for the disproportionately slow TS cleavage kinetics of AsCas12a R1226A actually lies in disproportionate slowing of a step prior to TS cleavage (i.e., NTS gap formation). Gap-widening cleavage events may be slower than the initial nicking event in this mutant due to the high entropic cost of associating a severed strand with the active site, as nicking of the NTS is expected to boost its conformational freedom (*Xiao et al., 2018*).

Similarly to Cas12a, Cas12i1-mediated TS cleavage depends upon NTS cleavage (*Appendix 2—figure 2—figure supplement 1*). However, cleavage-site mapping for Cas12i1 revealed that a NTS gap had already formed at the earliest timepoints for which cleavage was detectable (*Appendix 2—figure 2—figure supplement 11*). Additionally, Cas12i1-mediated TS cleavage could be only slightly accelerated by pre-gapping the NTS, indicating that NTS trimming is not rate-limiting for TS cleavage in Cas12i1 (*Appendix 2—figure 2B*, *Appendix 2—figure 2—figure supplement 8*). These results suggest that across diverse families of type V interference complexes, the microscopic steps of double-strand break formation can vary in absolute rate and relative kinetic breakdown.

Still, while both AsCas12a R1226A and Cas12i1 exhibit $k_{NTS}$:$k_{TS}$ ratios favorable for 'nickase' applications, their use may be limited by their low NTS cleavage rate, which is $10^2$–$10^3$ times slower than that of WT AsCas12a under the tested conditions. Thus, Cas9 remains the tool of choice for generating RNA-guided nicks in either DNA strand because each of its DNase domains can be independently inactivated by point mutation (*Jinek et al., 2012*).

