## [Decision Letter]

**Acceptance summary:**

This paper answers the puzzling question of how does a RNA guided CRISPR nuclease with a single active site (Cas12a) efficiently cleave both strands of its substrate DNA, an important question given Cas12a's use in gene editing? When Cas12a binds DNA the target strand base pairs with the guide RNA and the non-target strand is displaced forming an R-loop structure. It was found that this R-loop structure specifically destabilizes the double-stranded DNA 3' of the R-loop and that Cas12a the exploits this destabilized DNA to cut the second strand.

**Decision letter after peer review:**

Thank you for submitting your article "CRISPR-Cas12a exploits R-loop asymmetry to form double-strand breaks" for consideration by *eLife*. Your article has been reviewed by three peer reviewers, and the evaluation has been overseen by a Reviewing Editor and Cynthia Wolberger as the Senior Editor. The following individuals involved in review of your submission have agreed to reveal their identity: Ailong Ke (Reviewer #1); Malcolm F White (Reviewer #2).

The reviewers have discussed the reviews with one another and the Reviewing Editor has drafted this decision to help you prepare a revised submission.

Summary:

This manuscript by Cofsky et al. explores the mechanistic details underlying the poorly understood second strand cleavage catalysed by Cas12a – a type V CRISPR effector complex with a single RuvC active site. The crux of the question is to gain a molecular understanding for the observed activity of Cas12a, which cleaves the non-target strand and then the distorted target strand in the same active site. This question lies at the heart of the utility of Cas12a for genome engineering by targeted DSB formation.

The authors first use the classical method of Permanganate probing to assess the propensity of DNA bound by dCas12a to adopt a single stranded structure, identifying a region 3' of the R-loop that has a ss nature – an area coincident with the target site for Cas12a cleavage (Figure 2). They go on to demonstrate that the position of the R-loop junction dictates the site of target strand cleavage by Cas12a (Figure 3), and that R-loops specifically destabilise 3' R-loop flanks in a protein independent manner (Figure 4). This remarkable observation is confirmed using model nucleic acid structures (dumbbells), and appears to be due to differences in interhelical stacking (Figure 5).

Overall, this study reveals significant differences in the stability of R-loop:DNA junctions, irrespective of protein. On the one hand, this is a very interesting observation that could have widespread implications in nucleic acid biology. On the other, it remains unproven that the fundamental stability differences observed here have strong implications for the mechanism of Cas12a, as the data shown correlation but not necessarily a causative link between DNA junction stability and target strand cleavage. After all, nucleases such as Cas12a are quite capable of using binding energy to manipulate nucleic acid substrates to a remarkable degree, and the Cas12a second strand cleavage reaction requires considerable more than merely fraying of the R-loop junction.

That said, this is an interesting manuscript presenting a number of carefully designed experiments that yield some important new data with potentially wide relevance. The significance of the work is discussed in a thoughtful way and the paper will represent an impactful contribution to the field. We propose an essential revision that does not require new experiments.

Essential revision:

Section about "difference in interhelical stacking energy may underlie asymmetric R-loop flank stability": Here the authors "hypothesize that the asymmetry may emerge from energetic differences in the coaxial stacking of a DNA homoduplex on either end of an RNA:DNA hybrid", which got me lost. The authors refer to crystal structures of the Cas12a/R-loop structure in the 2019 Swarts and Jinek paper (PDB 6I1K and 6I1L). 6I1K best depicts the Cas12a/R-loop, however, the PAM-distal DNA duplex is not coaxially stacked underneath the DNA/RNA hybrid in the structure, it is rotated 180° to the side. I am not sure whether the discussion in this section and the molecular dynamics simulations in Figure 5 can directly explain the fraying propensity in PAM-distal DNA. Perhaps the authors should consider the possibility that the twisting of the backbone at the R-loop junction drives the DNA unwinding, which involves base-flipping in a sequential fashion from the junction of the R-loop. This rotation may be easier from the 3'-end of the R-loop because the backbone has a higher degree of rotation freedom, which means lower energetic barrier.

---

## [Author Response]

[…] Overall, this study reveals significant differences in the stability of R-loop:DNA junctions, irrespective of protein. On the one hand, this is a very interesting observation that could have widespread implications in nucleic acid biology. On the other, it remains unproven that the fundamental stability differences observed here have strong implications for the mechanism of Cas12a, as the data shown correlation but not necessarily a causative link between DNA junction stability and target strand cleavage. After all, nucleases such as Cas12a are quite capable of using binding energy to manipulate nucleic acid substrates to a remarkable degree, and the Cas12a second strand cleavage reaction requires considerable more than merely fraying of the R-loop junction.

We agree that protein binding energy is probably mechanistically important, and we agree that our data have not definitively established a causative link between DNA junction stability and target-strand cleavage. Instead, our data identify an interesting structural feature of CRISPR interference complexes that sets them apart from other nucleases that *only* have protein binding energy at their disposal. To catalyze target-strand cleavage, Cas12a likely relies on a combination of protein:DNA interactions and the native conformational dynamics of the nucleic acids themselves. We have made a minor adjustment to the final paragraph of the Concluding Remarks to clarify this point.

That said, this is an interesting manuscript presenting a number of carefully designed experiments that yield some important new data with potentially wide relevance. The significance of the work is discussed in a thoughtful way and the paper will represent an impactful contribution to the field. We propose an essential revision that does not require new experiments.Essential revision:Section about "difference in interhelical stacking energy may underlie asymmetric R-loop flank stability": Here the authors "hypothesize that the asymmetry may emerge from energetic differences in the coaxial stacking of a DNA homoduplex on either end of an RNA:DNA hybrid", which got me lost. The authors refers to crystal structures of the Cas12a/R-loop structure in the 2019 Swarts and Jinek paper (PDB 6I1K and 6I1L). 6I1K best depicts the Cas12a/R-loop, however, the PAM-distal DNA duplex is not coaxially stacked underneath the DNA/RNA hybrid in the structure, it is rotated 180° to the side. I am not sure whether the discussion in this section and the molecular dynamics simulations in Figure 5 can directly explain the fraying propensity in PAM-distal DNA. Perhaps the authors should consider the possibility that the twisting of the backbone at the R-loop junction drives the DNA unwinding, which involves base-flipping in a sequential fashion from the junction of the R-loop. This rotation may be easier from the 3'-end of the R-loop because the backbone has a higher degree of rotation freedom, which means lower energetic barrier.

We respond to these points individually here:

References to the crystal structures:

– We agree that this line of discussion is confusing, as the crystal structures do not serve as direct evidence for or against the mechanism that we discuss here. To clarify the main point of this section, we have removed the discussion of the crystal structures flagged by the reviewer. We have also removed some of the more detailed discussion at the end of this section that, retrospectively, seems extraneous to the main point. We thank the reviewer for identifying the weaknesses in this section, and we hope our changes have clarified our point.

Relationship of fraying propensity and stacking energy:

– Fraying is expected to occur more readily from a duplex terminus that is not stacked on top of another duplex (Häse and Zacharias, 2016). Therefore, interhelical junctions that spend less time in a stacked conformation (i.e., those with weaker stacking energy) are likely to exhibit a greater degree of fraying from each constituent duplex.

Alternative explanations for the observed DNA fraying behavior:

– There could be several explanations for the difference in fraying propensity at the two boundary types, and the “stacking energy hypothesis” presented in this manuscript is just one hypothesis that is supported by our experiments and simulations. We have added a sentence to the end of this section to indicate to the reader the level of certainty that we can currently attribute to our hypothesis.